# SurfelSplat: Learning Efficient and Generalizable Gaussian Surfel Representations for Sparse-View Surface Reconstruction

**Chensheng Dai,**[*] **Shengjun Zhang,**[*] **Min Chen, Yueqi Duan**[†]
Tsinghua University
{dcs23, zhangsj23,cm22}@mails.tsinghua.edu.cn, duanyueqi@tsinghua.edu.cn

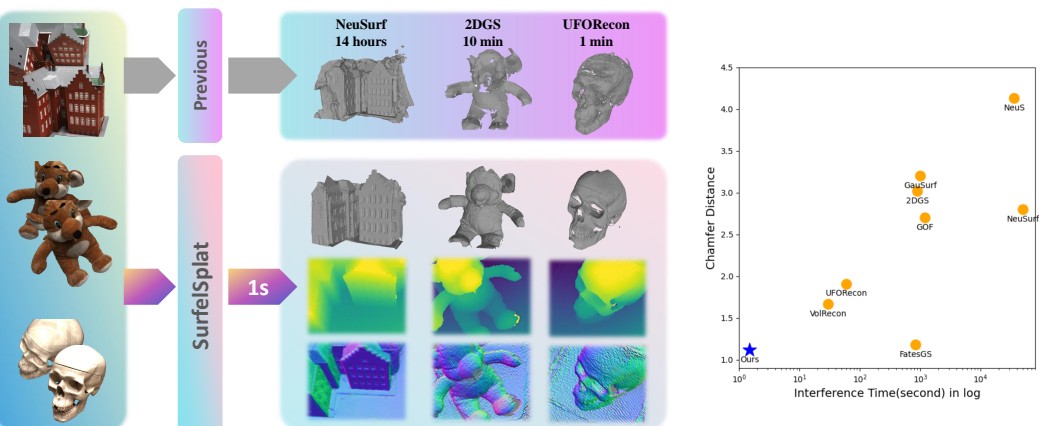

(a) Visualization and performance of our framework    (b) Performance and runtime comparison

Figure 1: Our method delivers state-of-the-art surface reconstruction with ultra-fast inference speed. (a) Framework visualization: Given an image pair, our approach regresses Gaussian radiance fields capturing fine geometric details in just 1 second. (b) Quantitative comparisons: Our method achieves superior reconstruction accuracy while maintaining the fastest runtime among existing approaches.

## Abstract

3D Gaussian Splatting (3DGS) has demonstrated impressive performance in 3D scene reconstruction. Beyond novel view synthesis, it shows great potential for multi-view surface reconstruction. Existing methods employ optimization-based reconstruction pipelines that achieve precise and complete surface extractions. However, these approaches typically require dense input views and high time consumption for per-scene optimization. To address these limitations, we propose SurfelSplat, a feed-forward framework that generates efficient and generalizable pixel-aligned Gaussian surfel representations from sparse-view images. We observe that conventional feed-forward structures struggle to recover accurate geometric attributes of Gaussian surfels because the spatial frequency of pixel-aligned primitives exceeds Nyquist sampling rates. Therefore, we propose a cross-view feature aggregation module based on the Nyquist sampling theorem. Specifically, we first adapt the geometric forms of Gaussian surfels with spatial sampling rate-guided low-pass filters. We then project the filtered surfels across all input views to obtain cross-view feature correlations. By processing these correlations through a

---

[*]Equal contribution
[†]Corresponding author.

39th Conference on Neural Information Processing Systems (NeurIPS 2025).

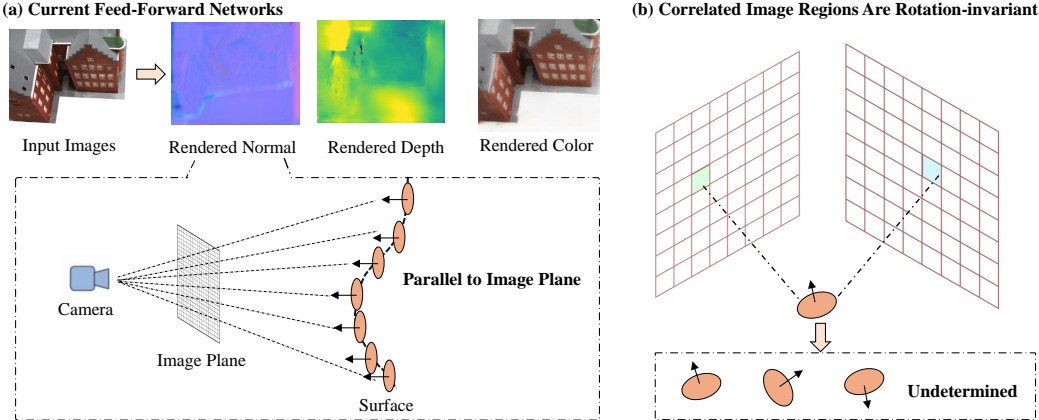

Figure 2: Experimental Observation. (a) Current feed-forward networks generate geometrically inaccurate Gaussian radiance fields. (b) The correlated image regions of pixel-aligned Gaussian surfels exhibit rotation invariance, limiting the network's ability to accurately infer surface orientations.

specially designed feature fusion network, we can finally regress Gaussian surfels with precise geometry. Extensive experiments on DTU reconstruction benchmarks demonstrate that our model achieves comparable results with state-of-the-art methods, and predict Gaussian surfels within 1 second, offering a 100× speedup without costly per-scene training.

# 1   Introduction

Reconstructing accurate surfaces from multi-view images remains a fundamental challenge in computer vision. Previous methods focus on Multi-View Stereo [1, 2] techniques to capture geometric details from multi-view images. Recent advancements of neural implicit representations [3, 4, 5, 6, 7, 8, 9, 10, 11, 12, 13] have demonstrated significant progress in recovering smooth and complete surface. However, these approaches typically struggle to extract precise surfaces in terms of sparse viewpoints. While following works [6, 7, 14, 15] have shown promise in sparse-view reconstruction, they generally require per-scene optimization with high time consumption. More recently, 3D Gaussian Splatting (3DGS) [16] has recently drawn increasing attention due to its rapid rendering speed and high visual fidelity. To enhance the surface alignment capabilities of Gaussian primitives, recent approaches [17, 18, 19, 20, 21] have modified the geometric shape of Gaussian representations to better conform to actual surfaces. For instance, 2D Gaussian Splatting (2DGS) [18] transforms 3D Gaussian primitives into 2D Gaussian surfels to maintain improved view-consistent geometry. While 3DGS-based methods succeed in precise surface extraction, they tend to overfit to the camera when presented with limited viewpoint information (*i.e.*, as few as two images), resulting in geometric collapse.

To circumvent per-scene optimization while ensuring generalizable and efficient scene reconstruction, several feed-forward networks [22, 23, 24, 25, 26, 27, 28, 29] have been proposed to directly regress 3D Gaussian parameters from sparse-view input images. These approaches predict the depth map and appearance attributes of pixel-aligned Gaussian primitives from cross-view image features. Current feed-forward frameworks achieve superior performance in fast and generalizable scene reconstruction for novel view synthesis. Therefore, an intuitive approach is to apply the current feed-forward networks for parameter prediction of 2D Gaussian surfels. However, as shown in Figure 2, typical methods such as MVSplat [24] fail to generate surfels with accurate geometry, where the normal vectors of surfels cannot be precisely recovered. The Gaussian surfels tend to orient parallel to the image plane rather than aligning with the actual surface geometry. As shown in Figure 2(b), Gaussian surfels predicted by these networks only cover the area of a single pixel. Consequently, the corresponding image regions relevant to surfel attributes cannot provide sufficient supervisory information to accurately learn the covariance of Gaussian surfels.

In this paper, we first analyze this phenomenon from the perspective of the Nyquist sampling theorem. Our key insight is that the failure to generate surface-aligned primitives is because the spatial frequency of pixel-aligned Gaussian surfels exceeds the Nyquist sampling rate, thus violating the fundamental signal processing principles. To trackle this challenge, we introduce SurfelSplat, a novel feed-forward framework to regress 2D Gaussian radiance field with precise geometry guided by Nyquist theorem. Our method dynamically modulates the geometric forms of diverse Gaussian surfels in the frequency domain and correlates pixel regions across multiple input views that effectively contribute to Gaussian geometric feature learning. We subsequently develop a feature aggregation network that leverages image features from these identified regions to enhance the original Gaussian image features, thereby yielding accurate Gaussian surfel representations with improved geometric fidelity. Our contributions are summarized as follows:

- We propose SurfelSplat, a feed-forward framework that regresses 2D Gaussian surfels directly from sparse-view images for surface reconstruction.

- We conduct a detailed analysis of why current feed-forward frameworks fail to generate geometrically-accurate Gaussian primitives and introduce Nyquist theorem-guided Gaussian surfel adaptations and feature aggregations to achieve superior geometric properties of the scene.

- Experimental results demonstrate the effectiveness of our method. SurfelSplat generates surface-aligned Gaussian radiance fields with high efficiency and accurate geometry.

## 2 Related Work

### 2.1 Neural Implicit 3D Representation

Neural Radiance Fields (NeRF) represent scenes through implicit radiance fields, with optimization processes dependent on volumetric rendering [30, 31, 32, 33, 34, 35, 36, 37, 38, 39, 40, 41, 42, 43, 44, 45, 46, 47, 48, 49]. For surface reconstruction, NeuS [3] pioneered scene representation using implicit Signed Distance Functions (SDFs) [5, 11, 12, 50]. The inherent continuity of MLP-based SDFs ensures smooth and accurate extracted meshes. Subsequent research has enhanced performance in sparse-view settings: VolRecon [51] integrates multi-scale feature extraction with geometry-aware regularization to recover 3D surfaces from limited viewpoints; NeuSurf [6] combines differentiable rendering with adaptive surface extraction techniques, enabling high-fidelity recovery of complex geometries; and UFORecon [8] employs an uncertainty-aware fusion optimization framework that leverages probabilistic feature correspondence and adaptive confidence weighting for robust surface reconstruction. However, the inherent complexity of volumetric rendering typically requires several hours of computation per scene.

### 2.2 Neural Explicit 3D Representation

Beyond neural implicit representations, 3D Gaussian Splatting (3DGS) has achieved remarkable progress in 3D scene reconstruction, delivering photorealistic rendering quality with high rendering speed [16, 52, 53, 54, 55, 56, 57, 58]. Two primary approaches have emerged for accurate surface extraction. The first enhances primitives to better fit surfaces: SuGaR [17] models 3D Gaussians as 2D pieces by incorporating flat and signed-distance regularization terms; 2DGS [18] and Gaussian Surfels [19] transform 3D Gaussian primitives into 2D surfels, with 2DGS proposing depth and normal consistency constraints to align surfels more accurately with surfaces. The second approach combines implicit representations to guide 3DGS training: NeuSG [59] integrates NeRF and 3DGS to recover complex 3D surfaces through differentiable optimization that preserves both local details and global structure; GSDF [21] employs a two-branch framework to simultaneously optimize SDF and Gaussian fields, allowing mutual enhancement to capture better geometric details. However, these methods require dense views to obtain smooth and complete surfaces due to the lack of scene data priors.

### 2.3 Generalizable Feed-forward Networks

The aforementioned optimization-based approaches have demonstrated strong performance in 3D reconstruction tasks, yet they typically require expensive per-scene training. More recently, feed-forward networks have emerged as a promising paradigm for generalizable 3D scene reconstruction.

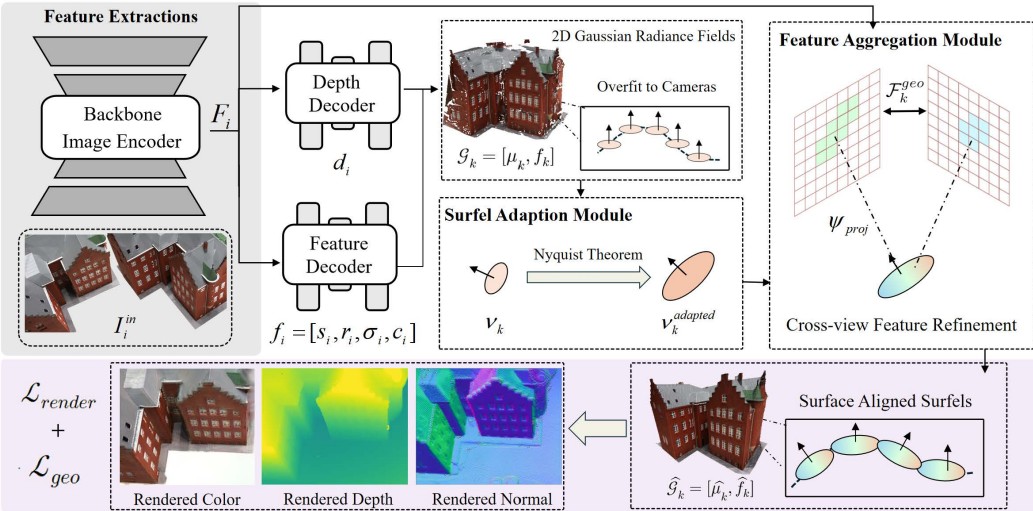

Figure 3: Pipeline. Given an image pair, our method first extracts initial image features using a backbone image encoder. We then predict Gaussian features and depth maps of the scene. Since 2D radiance fields are geometrically inaccurate, we apply Nyquist theorem-guided surfel adaptation to each surfel. In the feature aggregation module, we project the adapted surfels across views to identify image regions containing relevant geometric information. After refining the image features with these related regions, we regress the Gaussian radiance fields again to obtain accurate representations.

These models learn rich priors from large-scale datasets, enabling the reconstruction process to be accomplished through a single feed-forward inference. pixelNeRF [47] pioneered a feature-based framework that leverages encoded features to render novel views. Building upon Gaussian primitives as the fundamental representation, Splatter Image [22] and GPS-Gaussian [53] have achieved notable progress by predicting Gaussian parameters for object-level reconstruction. pixelSplat [23] further advanced this direction by regressing pixel-aligned Gaussian primitives, effectively incorporating epipolar geometry and depth estimation. MVSplat [24] enhances geometric quality by extracting cost volumes as cross-view features, which facilitates fast and accurate depth prediction. However, existing feed-forward methods predominantly target 3D reconstruction tasks such as novel view synthesis. Their potential for surface reconstruction—where significantly higher precision in Gaussian primitives is required—remains largely unexplored.

## 3 Methods

We present SurfelSplat, a feedforward framework for predicting 2D Gaussians with accurate geometric reconstruction, principled by the Nyquist sampling theorem (Figure 3). Our approach begins by predicting Gaussian centers and their associated attributes, followed by a surfel adaptation module that optimizes Gaussian primitives in the frequency domain. We then introduce a feature aggregation module that refines Gaussian representations by exploiting cross-view feature correlations. These refined features are subsequently utilized to regress surface-aligned 2D Gaussian radiance fields. In Section 3.1, we present a comprehensive analysis of spatial frequency characteristics and sampling rates for Gaussian surfels. Section 3.2 details our Nyquist-guided surfel adaptation and image feature aggregation modules, along with their corresponding network architectures. Our SurfelSplat is illustrated in Algorithm 1.

### 3.1 Sensitivity to Sampling Rates for Pixel-Aligned Gaussian Surfels

To overcome the limitations inherent in current pixel-aligned feedforward approaches, we initially examine the spatial sampling frequency within multi-camera systems and establish a methodology for computing the spatial frequency of individual 2D Gaussian primitives.

---

**Algorithm 1** SurfelSplat: Nyquist Sampling-Guided Gaussian Feature Aggregation

---

1: **Input:** Multi-view images $\mathcal{I} = \{\mathbf{I}_i\}_{i=1}^N$, camera parameters $\mathcal{P} = \{\mathbf{P}_i\}_{i=1}^N$, $f_x$ and $f_y$
2: **Output:** Gaussian parameters $\{\mu_k, \mathbf{r}_k, \mathbf{s}_k, \sigma_k, \mathbf{c}_k\}$ for each primitive $k$
3: $\mathcal{F} \leftarrow \Phi_{image}(\mathcal{I}, \mathcal{P})$
4: $\mu_i \leftarrow \psi_{unproj}(\Phi_{depth}, \mathbf{P}_i), \quad f_i \leftarrow \Phi_{attr}(F_i) \quad i = 1, 2, \cdots, N$
5: **for** $i \leftarrow 1$ to $N$ **do**
6: $\quad d_k^i \leftarrow \psi_{proj}(\mathcal{G}_k, \mathbf{P}_i)$
7: $\quad \hat{\nu}_k^i \leftarrow \frac{f_x f_y}{(d_k^i)^2}$
8: **end for**
9: $\hat{\nu}_k \leftarrow \max_i \hat{\nu}_k^i$
10: $\mathcal{G}_k^{low} \leftarrow exp(-\frac{\hat{\nu}_k^2 u^2}{2s^2} - \frac{\hat{\nu}_k^2 v^2}{2s^2}), \quad \hat{\mathcal{G}}_k^{adapted} \leftarrow \mathcal{G}_k \otimes \mathcal{G}_k^{low}$
11: $\mathcal{F}_k^{geo} \leftarrow \psi_{proj}(\mathcal{F}, \mathcal{P}, \hat{\mathcal{G}}_k^{adapted})$
12: $\mathbf{F}_k^{refined} \leftarrow \Phi_{refine}(\mathcal{F}_k^{geo}) + \mathbf{F}_k$
13: $\hat{\mathbf{f}}_k \leftarrow [\hat{\mathbf{s}}_k, \hat{\mathbf{r}}_k, \hat{\sigma}_k, \hat{\mathbf{c}}_k] = \Phi_{attr}(\mathbf{F}_k^{refined})$

---

### 3.1.1 Nyquist Sampling Theorem

The Nyquist Sampling Theorem [60] represents a cornerstone principle in signal processing. For precise reconstruction of a continuous signal from its discrete samples, the following criteria must be met:

**Nyquist Conditions** *The continuous signal must be band-limited with bandwidth $\nu$, and the spatial sampling rate $\hat{\nu}$ must be at least twice the signal bandwidth: $\hat{\nu} \geq 2\nu$.*

The Nyquist sampling theorem establishes the fundamental relationship between spatial signals and their corresponding sampling frequencies. In this work, we exploit the Nyquist criterion to learn local image features that significantly improve the reconstruction of fine-grained geometric scene details.

### 3.1.2 Spatial Sampling Rates in Multi-Camera Systems

For a single-camera system, the sampling interval in the image plane is determined by the pixel area. When projected into 3D space, this sampling interval corresponds to the area occupied on the surface manifold. For an image with focal lengths $f_x$ and $f_y$ (expressed in pixel units), the sampling interval in screen space is unity. Consider a unit area element $dA_{xy}$ in screen space and its corresponding surface area coverage $dA_{uv}$. The sampling rate in 3D space can then be derived as:

$$\hat{\nu}_{sampling} = \frac{dA_{xy}}{dA_{uv}} \tag{1}$$

The relationship between these two parameter spaces is given by $dA_{xy} = |\mathbf{J}|du \cdot dv$, where $|\mathbf{J}|$ represents the determinant of Jacobian matrix: $\mathbf{J} = \frac{\partial \mathbf{P}_{image}(x,y)}{\partial \mathbf{X}_{camera}} \cdot \frac{\partial \mathbf{X}_{camera}}{\partial \mathbf{X}_{world}} \cdot \frac{\partial \mathbf{X}_{world}}{\partial (u,v)}$. By evaluating the spatial projection relationship that governs the projection process, we obtain the sampling frequency for a given spatial primitive:

$$\hat{\nu}_{sampling} = |\mathbf{J}| = \frac{f_x f_y}{d^2} \tag{2}$$

where $d$ denotes the corresponding depth value. The detailed mathematical derivation is provided in the Appendix B.1. For a multi-camera system, the spatial sampling frequency is computed across all cameras. We define the overall sampling frequency for a Gaussian primitive $p_k$ as the maximum frequency among all views:

$$\hat{\nu}_k = \max \left( \{\mathbb{V}_i(p_k) \cdot |J_i|\}_{i=1}^N \right) \tag{3}$$

where $N$ represents the number of cameras and $\mathbb{V}_i$ denotes the visibility function. If the primitive is visible to the $i$-th camera, $\mathbb{V}_i$ returns 1, otherwise 0. Specifically, our choice of using the maximum sampling frequency as the overall frequency (Equation 3) is motivated by Equation 7 of Mip-Splatting [54]. The key insight of Mip-Splatting is that for accurate reconstruction, we need to ensure that each

3D Gaussian primitive satisfies the Nyquist sampling criterion for **at least one camera view** where it is visible. This is because if a primitive can be accurately reconstructed from at least one view, we have captured its essential geometric information.

### 3.1.3 Spatial Frequency of 2D Gaussian Primitives

Given a spatial surfel, the spatial frequency can be calculated through spatial Fourier transform derivation $|\hat{G}(\mathbf{k})|$. Since the Gaussian function contains over 95% of its energy within $\pm 2$ standard deviations, when considering a Gaussian with two standard deviations as the surfel size, we can obtain the frequencies along the tangent vector directions $\mathbf{t}_u$ and $\mathbf{t}_v$ in the 2D Gaussian surfel via $|\hat{G}(\mathbf{t}_u)|$ and $|\hat{G}(\mathbf{t}_v)|$, respectively. The detailed derivation can be found in Appendix B.2.

Consequently, along the $\mathbf{t}_u$ direction, the frequency is $\omega_u = \frac{2}{s_u}$ (and analogously, $\omega_v = \frac{2}{s_v}$ for the $\mathbf{t}_v$ direction). Accounting for the $2\pi$ periodic normalization of the Fourier transform, the spatial frequency of the Gaussian primitive along each tangent vector can be expressed as:

$$\nu_u = \frac{1}{\pi s_u}, \quad \nu_v = \frac{1}{\pi s_v} \tag{4}$$

For Gaussian primitives that fail to satisfy the Nyquist criterion, the spatial signal cannot be perfectly reconstructed. In such cases, the network tends to predict spatial parameters (e.g., covariance) with considerable stochasticity, resulting in surfels that are misaligned with the actual surface geometry.

## 3.2 Surfel Prediction with Nyquist Theorem-Guided Feature Aggregation

Having established the methodology for calculating sampling rates and spatial primitive frequencies, we proceed to design modules that enable Gaussian primitive predictions to adhere to the Nyquist sampling criterion. Specifically, we perform Gaussian surfel adaptation in the frequency domain and employ cross-view feature aggregation to regress primitives with enhanced geometric detail fidelity.

### 3.2.1 Nyquist Theorem-Guided Gaussian Surfel Adaptation

We aim to constrain the maximum frequency of $\mathcal{G}_k$ according to the spatial sampling rates. We propose an adaptive surfel adaptation module operating in the frequency domain. Specifically, we achieve this by passing 2D Gaussian primitives through an adaptive Gaussian low-pass filter:

$$\hat{\mathcal{G}}_k^{\text{adapted}}(\mathbf{x}) = (\mathcal{G}_k \otimes \mathcal{G}_k^{\text{low}})(\mathbf{x}), \quad \mathcal{G}_k^{\text{low}}(x) = e^{-\frac{\hat{\nu}_k^2 u^2}{2s^2} - \frac{\hat{\nu}_k^2 v^2}{2s^2}} \tag{5}$$

Here, $s$ is a scalar hyperparameter (default value is 1), and each Gaussian filter is designed according to the specific frequency bound $\hat{\nu}_k$. We then adaptively modify the transformation matrix of the 2D Gaussian primitive $\mathbf{H}_k^{\text{adapted}}$ as the scaling matrix changes:

$$\mathbf{H}_k^{\text{adapted}} = \begin{bmatrix} s_u \sqrt{1 + \frac{s^2}{\hat{\nu}_k^2}} \mathbf{t}_u & s_v \sqrt{1 + \frac{s^2}{\hat{\nu}_k^2}} \mathbf{t}_v & \mathbf{0} & \mathbf{p}_k \\ 0 & 0 & 0 & 1 \end{bmatrix} = \begin{bmatrix} \mathbf{RS}_k^{\text{adapted}} & \mathbf{p}_k \\ \mathbf{0} & 1 \end{bmatrix} \tag{6}$$

where $\mathbf{S}_k^{\text{adapted}}$ is the adapted scaling matrix, and the transformation matrix $\mathbf{H}_k^{\text{adapted}}$ completely characterizes the 2D Gaussian representation, incorporating the effects of the low-pass filter.

**Theoretical Nyquist Criterion Verification**  Prior to adaptation, whether all primitives satisfy the Nyquist criterion cannot be determined. After adaptation, the spatial frequency can be constrained by setting $s_u > \frac{2}{\pi}$:

$$\nu_k = \frac{1}{s_u \pi \sqrt{1 + \frac{1}{\hat{\nu}_k^2}}} < \frac{\hat{\nu}_k}{s_u \pi} < \frac{\hat{\nu}_k}{2} \tag{7}$$

Regardless of how the spatial sampling rates vary, the Nyquist criterion is consistently satisfied.

### 3.2.2 Nyquist Theorem-Guided Gaussian Feature Aggregation

**Gaussian Parameters Initialization**  Given N input images $\mathcal{I} = \{\mathbf{I}_i\} \in \mathbb{R}^{N \times H \times W \times 3}$ and corresponding camera parameters $\mathcal{P} = \{\mathbf{P}_i\}, \mathbf{P}_i = \mathbf{K}_i[\mathbf{R}_i|\mathbf{t}_i]$, we first use epipolar transformers to

extract rough image features, and use cost volumes between perspective pairs to extract geometric interrelationships. We then concatenate these features to obtain our initial image features:

$$\mathcal{F} = \Phi_{initial}(\mathcal{I}), \mathcal{F} = \{\mathbf{F}_i\} \in R^{N \times W \times H \times C} \tag{8}$$

where $\Phi_{initial}$ is the image feature extraction backbone. In conventional feedforward frameworks, cross-view features are fed into two distinct regression networks $\Phi_{depth}$ and $\Phi_{attr}$ to predict depth $\mathbf{d}_i$ and Gaussian attributes $\mathbf{f}_i = [\mathbf{s}_i, \mathbf{r}_i, \sigma_i, \mathbf{c}_i]$:

$$\mathbf{d}_i = \Phi_{depth}(\mathbf{F}_i) \in \mathbb{R}^{HW}, \mu_i = \psi_{unproj}(\mathbf{d}_i, \mathbf{P}_i) \in \mathbb{R}^{HW \times 3}, \mathbf{f}_i = \Phi_{attr}(\mathbf{F}_i) \in \mathbb{R}^{HW \times C_{attr}} \tag{9}$$

where $\psi_{unproj}$ denotes the unprojection process.

**Cross-view Gaussian Feature Aggregation**  Given the frequency distribution in space, we perform Gaussian surfel adaptations for each primitive $\hat{\mathcal{G}}_k^{\text{adapted}}(\mathbf{x}) = (\mathcal{G}_k \otimes \mathcal{G}_k^{\text{low}})(\mathbf{x})$. Within our framework, we project 2D Gaussian primitives back to all viewpoints to extract the set of image features required for refinement. With lower frequency, primitives tend to occupy more pixels related to Gaussian attributes regression. The image regions $\mathcal{R}_k$ associated with $\hat{\mathcal{G}}_k^{\text{adapted}}$ are defined by:

$$\mathcal{R}_k^i = \{\mathbf{x} = (i,j) \in \mathbb{Z}^2 : \hat{\mathcal{G}}_k^{\text{adapted}}(i,j;m) > \epsilon\}, \quad \mathcal{R}_k = \bigcup_{i=1}^{N} \mathcal{R}_k^i \tag{10}$$

where $\hat{\mathcal{G}}_k^{\text{adapted}}(i,j;m)$ represents the Gaussian value of primitive $\hat{\mathcal{G}}_k^{\text{adapted}}$ splatted onto the $m^{\text{th}}$ view at pixel $(i,j)$.

We can then identify image features associated with the geometric information of primitive $\mathcal{G}_k$:

$$\mathcal{F}_{i,k}^{\text{geo}} = \{\mathbf{F}_k^i(i,j), (i,j) \in \mathcal{R}_k^i\}, \quad \mathcal{F}_k^{\text{geo}} = \bigcup_{i=1}^{N} \mathcal{F}_{i,k}^{\text{geo}} \tag{11}$$

**Gaussian Prediction with Refined Feature**  As features in $\mathcal{F}_k^{\text{geo}}$ are essential for accurate geometry learning of our Gaussian representation $\mathcal{G}_k$, we implement a feature refinement architecture with cross-attention transformations to enhance the initial image feature $\mathbf{F}_k$. The query, key, and value composition is specifically designed to enable cross-attention interaction for a Gaussian primitive $\mathcal{G}_k$ as $\hat{F}_k = \Phi_{\text{Att}}(Q, K, V)$:

$$Q = h_Q(\mathbf{F}_k), \quad K = h_K(\mathcal{F}_k^{geo}), \quad V = h_V(\mathcal{F}_k^{geo}) \tag{12}$$

We then employ a standard feed-forward architecture in the transformer:

$$\mathbf{F}_k^{\text{refined}} = \Phi_{\text{FFN}}(\hat{\mathbf{F}}_k) + \mathbf{F}_k \tag{13}$$

Finally, we predict geometry-aware pixel-aligned 2D Gaussian primitives with the refined feature per view $\mathbf{F}_i^{\text{refined}} = \{\mathbf{F}_k^{\text{refined}}, \mathcal{G}_k \subset \mathcal{I}_i\}$ using the same Gaussian head as in Equation 9:

$$\hat{\mathbf{f}}_i = [\hat{\mathbf{s}}_i, \hat{\mathbf{r}}_i, \hat{\sigma}_i, \hat{\mathbf{c}}_i] = \Phi_{\text{attr}}(\mathbf{F}_i^{\text{refined}}) \in \mathbb{R}^{HW \times C_{\text{attr}}} \tag{14}$$

### 3.3  Loss Design

Our loss function comprises two parts: rendering loss and geometric loss. The rendering loss $\mathcal{L}_{render} = \mathcal{L}_{RGB} + \lambda_{LPIPS}\mathcal{L}_{LPIPS}$ employs mean square error along with LPIPS loss. For geometric loss, we use depth and normal continuity functions to align surfels to the surface: $\mathcal{L}_{align} = \sum_i \omega_i(1 - n_i^T N)$. Furthermore, we incorporate depth and normal mean square error: $\mathcal{L}_{geo} = \lambda_{align}\mathcal{L}_{align} + \lambda_d\mathcal{L}_d + \lambda_n\mathcal{L}_n$. Our complete loss function is formulated as:

$$\mathcal{L} = \mathcal{L}_{render} + \lambda_{geo}\mathcal{L}_{geo} \tag{15}$$

## 4  Experiments

To demonstrate the effectiveness of our method, we conduct experiments on DTU benchmarks [61] and compare the reconstruction accuracy and evaluation efficiency with state-of-the-art methods. Additionally, we provide a detailed analysis of the geometric properties from the perspective of the Nyquist sampling criterion to further validate our approach. In the ablation study, we analyze the effectiveness of each component of our method.

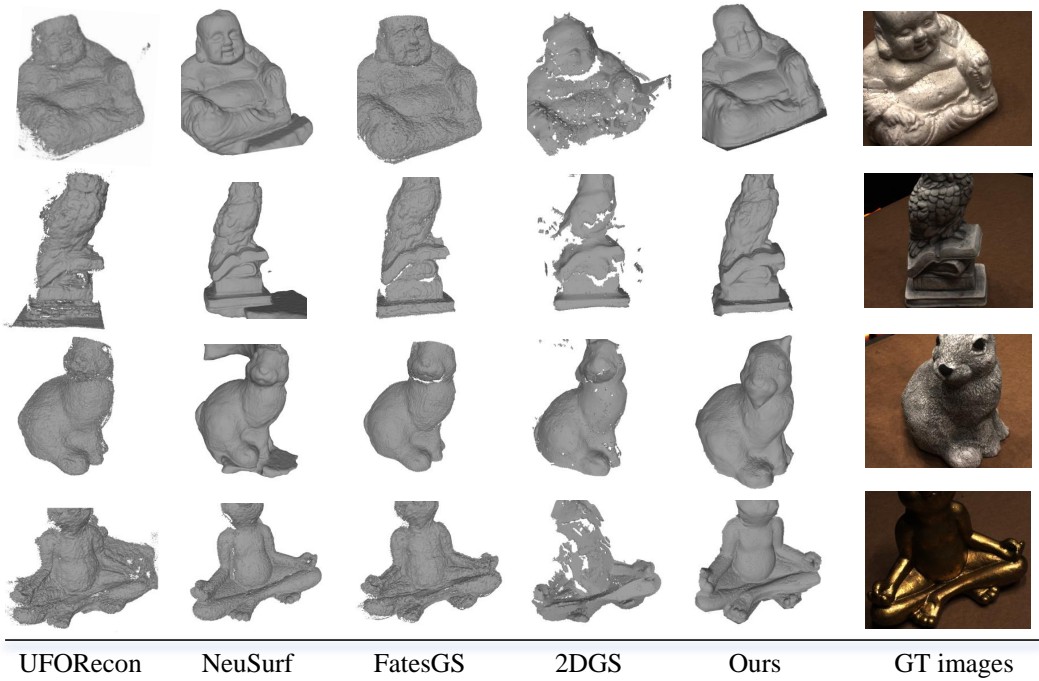

| UFORecon | NeuSurf | FatesGS | 2DGS | Ours | GT images |

Figure 4: Qualitative Comparison of Surface Reconstruction with Sparse Views on DTU Benchmarks.

## 4.1 Experimental Setup

**Datasets.**  We evaluate our method on the DTU dataset. DTU consists of 128 scenes, with 15 scenes designated for testing. We assess reconstruction accuracy using Destination to Source (D2S) Chamfer Distance as the evaluation metric. The investigated experimental setting is sparse-view reconstruction with 2 input views. Input images are downsampled to a resolution of $256 \times 320$ pixels.

**Implementation Details.**  Our implementation is built upon the pixelSplat [23] framework. The training process consists of two stages: first, we train our model on RealEstate10K [62] for 300,000 iterations, followed by fine-tuning on the DTU dataset for 2,000 iterations. The hyperparameter $s$ in Equation 5 is set to 1. All experiments reported in this paper were conducted on a single NVIDIA RTX A6000 GPU using the Adam optimizer.

## 4.2 Comparisons

**Sparse view surface reconstruction.**  As shown in Table 1, our SurfelSplat exhibits the best mean D2S reconstruction Chamfer distance (CD) performance compared to other state-of-the-art surface reconstruction methods. As illustrated in Figure 4, our method presents superior global geometry and exhibits enhanced surface details. In contrast to UFORecon [8], which can only produce coarse global geometry, our method demonstrates improved global surface smoothness. We can also refine local details that would be ignored by methods like 2DGS [18], which delivers coarse and incomplete surfaces. Additional experimental results on the BlendedMVS [63] dataset are presented in Appendix C.4.

**Efficiency.**  We conduct efficiency studies on all tested scenes for the sparse-view reconstruction methods mentioned above. As highlighted in Table 2, we compare the mean inference time on DTU benchmarks. All experiments are conducted on the same device.

For neural implicit methods that require per-scene training, convergence requires significantly long training times. Neural explicit training methods greatly reduce training time consumption, but still require approximately 10 minutes to obtain the Gaussian radiance fields. Most recent implicit methods have successfully compressed the inference time to the 1-minute level. However, our method shows the best efficiency with a single feed-forward process that takes only seconds.

Table 1: The quantitative comparison results of Chamfer Distance (CD↓) on DTU dataset. The best results are in **bold**, the second best are underlined.

| ID | 24 | 37 | 40 | 55 | 63 | 65 | 69 | 83 | 97 | 105 | 106 | 110 | 114 | 118 | 122 | Mean |
|---|---|---|---|---|---|---|---|---|---|---|---|---|---|---|---|---|
| NeuS [3] | 4.69 | 4.72 | 4.03 | 4.58 | 4.71 | 2.01 | 4.83 | 3.94 | 4.31 | 2.61 | 1.63 | 6.48 | 1.44 | 5.69 | 6.34 | 4.13 |
| NeuSurf [6] | 1.96 | 3.73 | 2.35 | **0.82** | **1.07** | 2.51 | 0.87 | 1.21 | 1.15 | 1.13 | 1.06 | 1.23 | **0.41** | 0.92 | 1.13 | 1.44 |
| VolRecon [51] | 1.41 | 3.24 | 1.76 | 1.43 | 1.66 | 2.25 | 1.42 | 1.81 | 1.54 | 1.26 | 1.52 | 1.53 | 0.99 | 1.54 | 1.75 | 1.67 |
| UFORecon [8] | 1.15 | 2.42 | 1.67 | 2.55 | 1.90 | 2.73 | 1.55 | 1.49 | 2.16 | 0.95 | 2.22 | 1.98 | 1.40 | 2.11 | 2.32 | 1.91 |
| 2DGS [18] | 2.29 | 2.63 | 2.33 | 1.23 | 3.69 | 4.71 | 2.64 | 3.94 | 3.55 | 3.92 | 3.95 | 2.68 | 2.37 | 3.15 | 2.21 | 3.02 |
| GausSurf [19] | 4.22 | 5.69 | 4.32 | 3.98 | 4.93 | 2.81 | 4.67 | 5.52 | 4.98 | 3.61 | 4.11 | 5.43 | 2.98 | 3.66 | 4.55 | 4.36 |
| FatesGS [64] | **0.77** | 2.35 | **1.43** | 1.00 | 1.31 | 2.06 | **0.85** | 1.24 | **1.06** | 0.83 | 1.22 | **0.58** | 0.64 | 0.99 | 1.32 | 1.18 |
| **Ours** | 1.23 | **1.69** | 1.63 | 0.90 | 1.24 | **1.14** | 1.12 | **1.18** | 1.13 | **0.79** | **0.84** | 1.02 | 0.98 | **0.84** | **1.04** | **1.12** |

Table 2: Comparisons with reconstruction efficiency.

| Method | Interference Time |
|---|---|
| NeuS | $10_{\pm 0.5}$ hours |
| NeuSurf | $14_{\pm 0.5}$ hours |
| VolRecon | $60_{\pm 5}$ seconds |
| UFORecon | $100_{\pm 5}$ seconds |
| 2DGS | $10_{\pm 0.5}$ minutes |
| GauSurf | $2_{\pm 0.2}$ hours |
| FatesGS | $14_{\pm 0.5}$ minutes |
| **Ours** | $1_{\pm 0.05}$ second |

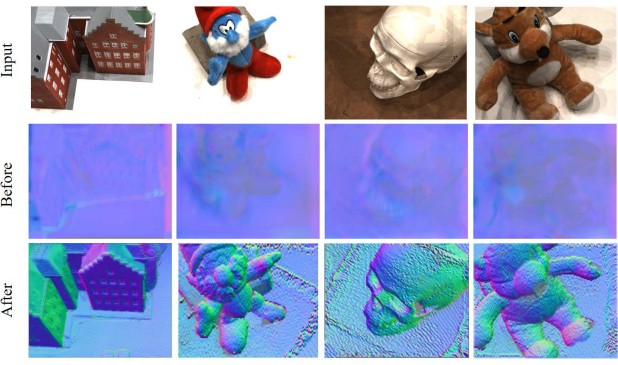

Figure 5: Visualization of Nyquist Theorem Verification

## 4.3 Experimental Nyquist Theorem Verifications

In Section 3.1, we analyze in detail how to derive the Nyquist sampling rates and spatial frequency of a Gaussian primitive. In our theoretical analysis, we prove that our surfel adaptation module can adjust the spatial frequency within Nyquist thresholds. To further demonstrate the effectiveness of our method, we conduct experiments on evaluated scenes for Nyquist criterion verification. We record the rendered depth maps and scale factor distributions from all tested scenes, and calculate the corresponding sampling rates and spatial frequencies. From the Nyquist criterion, we know that $\nu_k$ and $\hat{\nu}_{Nyquist} = \frac{\hat{\nu}_{sampling}}{2}$ must satisfy $\frac{\nu_{surfel}}{\hat{\nu}_{Nyquist}} < 1$, so we summarize the normalized frequency ratio $\frac{\nu_{surfel}}{\hat{\nu}_{Nyquist}}$ across all Gaussian surfels.

As illustrated in Figure 6, we can see that before surfel adaptation, almost all Gaussian primitives exceed the Nyquist threshold. The network cannot obtain sufficient information during the backpropagation stage and thus is unable to recover precise geometry. After the surfel adaptation module, all Gaussian primitives fall within the Nyquist frequency boundary. As shown in Figure 5, the rendered normal maps before and after the surfel adaptation module show significant differences, which further validates our method.

## 4.4 Ablation Studies

To demonstrate the necessity and effectiveness of our proposed components, we conducted experiments on DTU evaluation scenes to measure the impact of individual technical designs on reconstruction performance. The proposed modules are tested for ablation: the surfel adaptation module and the feature aggregation module. As shown in Table 3, in addition to the mean Chamfer Distance values, we also evaluate the normal rendering errors. For ground truth normal vectors, we utilize the normal maps provided by Gaussian Surfel [19]. We conduct experiments with 2 input views and render the normal maps on the same views. The results demonstrate that removing any of the proposed modules results in different performance degradation, confirming the effectiveness of each proposed component.

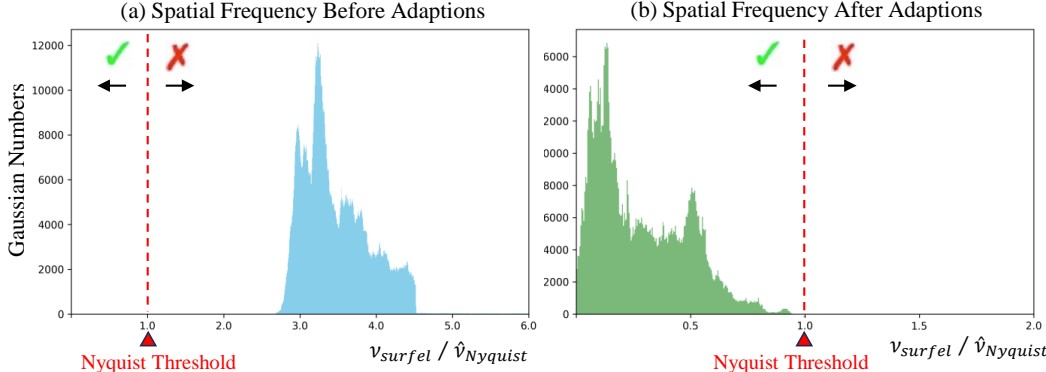

Figure 6: Nyquist Theorem Verification: (a) Before adaptation, most surfels exceed the Nyquist threshold, resulting in inaccurate geometry prediction. (b) After the adaptation module, all Gaussian primitives fall within the Nyquist threshold, ensuring accurate geometric feature learning.

Table 3: Ablation study of surfel adaption and feature aggregation module on DTU benchmarks.

| Method | CD↓ | Normal MSE↓ |
|---|---|---|
| w/o Adaption. | 2.56 | 0.135 |
| w/o Aggre. | 1.96 | 0.115 |
| **Ours** | **1.12** | **0.060** |

## 5    Conclusion and Discussion

In this paper, we propose SurfelSplat to predict surface-aligned Gaussian surfel representations from sparse-view images. To regress geometrically precise surfels, we apply Nyquist sampling criterion-guided surfel adaptation and feature aggregation modules to make the spatial frequency conform to the frequency constraints. Experimental results demonstrate that our method generates Gaussian radiance fields with more precise geometry and higher efficiency.

Although SurfaceSplat outperforms prior works, it has limitations. Since we predict pixel-aligned Gaussians for each view, the radiance fields are sensitive to image resolution. With higher resolutions such as $1024 \times 1024$, over 1 million Gaussian surfels would degrade both rendering and inference speed. Moreover, the unseen parts of the scene limit reconstruction performance, suggesting that generative models such as diffusion models could be introduced into our framework. Consequently, several promising directions remain to be explored.

We also acknowledge that the efficiency of our method benefits from the feed-forward architecture. However, integrating surface reconstruction effectively into feed-forward networks presents significant challenges: the orientations of Gaussian primitives cannot be correctly recovered due to insufficient spatial sampling frequency. To address this, we adopt surfel adaptation modules that enable each Gaussian primitive to acquire adequate geometric information, guided by the Nyquist sampling theorem, thereby achieving geometrically fine Gaussian radiance fields within the feed-forward framework.

## Acknowledgments

This work was supported in part by the Beijing Natural Science Foundation under Grant L252011, and by the National Natural Science Foundation of China under Grant 62206147.

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

# Appendix

## A  Preliminaries

### A.1  Surfels: Surface Elements

Surface elements, commonly referred to as *surfels*, constitute a point-based representation paradigm for modeling three-dimensional surfaces without explicit connectivity information. Originally introduced by Pfister et al. [65], surfels have emerged as a powerful alternative to traditional mesh-based representations in various surface reconstruction applications.

A surfel $s$ is formally defined as a tuple:

$$s = \mathbf{p}, \mathbf{n}, r, \mathbf{c}, \sigma \tag{16}$$

where:

- $\mathbf{p} \in \mathbb{R}^3$ denotes the 3D position vector of the surfel
- $\mathbf{n} \in \mathbb{S}^2$ represents the unit normal vector ($\mathbb{S}^2$ being the unit sphere in $\mathbb{R}^3$)
- $r \in \mathbb{R}^+$ specifies the radius (or size) of the surfel
- $\mathbf{c} \in \mathbb{R}^3$ or $\mathbb{R}^4$ encodes color information (optional)
- $\sigma \in \mathbb{R}^+$ indicates the confidence or uncertainty measure (optional)

Each surfel can be geometrically interpreted as a local surface approximation, typically visualized as a oriented disk centered at $\mathbf{p}$ with radius $r$ and orientation defined by $\mathbf{n}$. Collectively, a set of surfels $\mathcal{S} = s_1, s_2, ..., s_N$ forms a discrete sampling of the underlying continuous surface $\mathcal{M}$.

### A.2  2D Gaussian Splatting

2D Gaussian splatting (2DGS) has demonstrated remarkable efficacy in achieving accurate and smooth surface extraction. Each 2D Gaussian primitive represents a tangent plane in 3D space characterized by three key parameters: a central position $\mathbf{p}_k$, two orthogonal tangential vectors $\mathbf{t}_u$ and $\mathbf{t}_v$, and a scaling vector $\mathbf{s} = (s_u, s_v)$ that determines the covariance of the 2D Gaussian primitive.

The normal vector of the surfel is computed as $\mathbf{t}_w = \mathbf{t}_u \times \mathbf{t}_v$. The rotation matrix is defined as $\mathbf{R} = [\mathbf{t}_u, \mathbf{t}_v, \mathbf{t}_w] \in \mathbb{R}^{3 \times 3}$, while the scaling vector is arranged into a diagonal scaling matrix $\mathbf{S} \in \mathbb{R}^{3 \times 3}$ with its last diagonal entry set to zero.

A point $P$ in world space on the 2D Gaussian surfel $p_k$ is defined in the local tangent space and parameterized by:

$$P(u,v) = \mathbf{p}_k + s_u \mathbf{t}_u u + s_v \mathbf{t}_v v = \mathbf{H}(u, v, 1, 1)^T \tag{17}$$

$$\mathbf{H} = \begin{bmatrix} s_u \mathbf{t}_u & s_v \mathbf{t}_v & \mathbf{0} & \mathbf{p}_k \\ 0 & 0 & 0 & 1 \end{bmatrix} = \begin{bmatrix} \mathbf{RS} & \mathbf{p}_k \\ \mathbf{0} & 1 \end{bmatrix} \tag{18}$$

where $\mathbf{H} \in \mathbb{R}^{4 \times 4}$ denotes the homogeneous transformation matrix. For a point $\mathbf{u} = (u, v)$ on the tangent plane, its Gaussian value is defined by: $\mathcal{G}(\mathbf{u}) = \exp\left(-\frac{u^2 + v^2}{2}\right)$.

In this paper, we adopt the 2D Gaussian primitive as the fundamental surfel representation. The surfel attributes mentioned in Equation 16 can be mapped as follows: the center $\mathbf{p}$ corresponds to the central position $\mathbf{p}_k$; the normal vector is defined by $\mathbf{t}_w = \mathbf{t}_u \times \mathbf{t}_v$; the radius $r$ is represented by the scaling vector $\mathbf{s} = (s_u, s_v)$; and the color $\mathbf{c}$ and uncertainty $\sigma$ correspond to the spherical harmonic coefficients and opacity, respectively.

### A.3  Spatial Fourier Transform

The Spatial Fourier Transform (SFT) provides a framework for analyzing spatial frequency components in multidimensional signals. For a continuous function $f(\mathbf{x})$ where $\mathbf{x} \in \mathbb{R}^d$ represents spatial coordinates in a $d$-dimensional space, the SFT is defined as:

$$\mathcal{F}\{f(\mathbf{x})\} = F(\mathbf{k}) = \int_{\mathbb{R}^d} f(\mathbf{x}) e^{-i\mathbf{k} \cdot \mathbf{x}} d\mathbf{x} \tag{19}$$

where $\boldsymbol{\omega} \in \mathbb{R}^d$ denotes the spatial frequency vector and $i$ is the imaginary unit. Correspondingly, the inverse SFT is expressed as:

$$\mathcal{F}^{-1}\{F(\mathbf{k})\} = f(\mathbf{x}) = \int_{\mathbb{R}^d} F(\mathbf{k})e^{i\mathbf{k}\cdot\mathbf{x}}d\mathbf{k} \tag{20}$$

### A.4 Basic Assumptions on the surface manifold

In the main body of our paper, the real signal we aim to recover is the 3D surface manifold of the scene. However, we only have access to discrete 2D image observations of this continuous 3D signal. And 2D Gaussian surfels are our chosen representation to approximate this surface.

Specifically, we model the real signal using a collection of 2D Gaussian primitives (following the foundation established by 2DGS [18] and Gaussian Surfels [19]). The problem of reconstructing the surface from discrete 2D sampling is thus reformulated as reconstructing the Gaussian primitives from the 2D image data.

## B  Mathematical Derivations

### B.1  Spatial Sampling Rates in Multi-Camera Systems

In this paper, we provide a comprehensive derivation of the spatial sampling rates for a single-camera system. Intuitively, the spatial sampling interval in image space is unity, and the width and height of the sampling area are $\frac{f}{d}$ times larger than the spatial sampling interval of the image space. Therefore, we can readily derive the sampling frequency as $\frac{f_x f_y}{d^2}$.

Specifically, the spatial sampling rate $\hat{\nu}_{sampling}$ is determined by $\hat{\nu}_{sampling} = \frac{dA_{xy}}{dA_{uv}} = |\mathbf{J}|$, where $\mathbf{J}$ denotes the Jacobian matrix of the projection process. The projection process can be characterized by the following Jacobian matrix:

$$\mathbf{J} = \frac{\partial \mathbf{P}_{image}(x,y)}{\partial \mathbf{X}_{camera}} \cdot \frac{\partial \mathbf{X}_{camera}}{\partial \mathbf{X}_{world}} \cdot \frac{\partial \mathbf{X}_{world}}{\partial(u,v)} \tag{21}$$

First, a point in camera space $X_C$ is derived from its corresponding point in world space $X$ through the transformation: $X_C = RX + t$.

Subsequently, we project $X_C$ onto the image plane, establishing the relationship between the point on the image plane $\mathbf{x}$ and $X_C$:

$$\mathbf{x} = \begin{pmatrix} x \\ y \end{pmatrix} = \begin{pmatrix} f_x \frac{X_C}{Z_C} + c_x \\ f_y \frac{Y_C}{Z_C} + c_y \end{pmatrix} \tag{22}$$

The corresponding Jacobian matrix can then be calculated as:

$$J_{2\times3} = \frac{\partial \vec{x}}{\partial X} = \frac{\partial \vec{x}}{\partial X_C}\frac{\partial X_C}{\partial X} = \frac{1}{Z_C}\begin{pmatrix} f_x & 0 & -\frac{f_x}{Z_C}X_C \\ 0 & f_y & -\frac{f_y}{Z_C}Y_C \end{pmatrix} R \tag{23}$$

where $Z_C$ represents the depth value $D(x,y)$ at the corresponding image coordinates. Analogously, the inverse transformation yields:

$$J_{3\times2} = \frac{\partial X}{\partial X_C}\frac{\partial X_C}{\partial(u,v)} = R^T \frac{\partial X_C}{\partial(u,v)} = R^T \begin{pmatrix} 1 + \frac{u-c_x}{f_x}\frac{\partial Z_C}{\partial u} & \frac{u-c_x}{f_x}\frac{\partial Z_C}{\partial v} \\ \frac{v-c_y}{f_y}\frac{\partial Z_C}{\partial u} & 1 + \frac{v-c_y}{f_y}\frac{\partial Z_C}{\partial v} \\ \frac{\partial Z_C}{\partial u} & \frac{\partial Z_C}{\partial v} \end{pmatrix} \tag{24}$$

The overall Jacobian matrix is obtained through composition:

$$J = \frac{1}{Z_C}\begin{pmatrix} f_x & 0 & -\frac{f_x}{Z_C}X_C \\ 0 & f_y & -\frac{f_y}{Z_C}Y_C \end{pmatrix} RR^T \begin{pmatrix} 1 + \frac{u-c_x}{f_x}\frac{\partial Z_C}{\partial u} & \frac{u-c_x}{f_x}\frac{\partial Z_C}{\partial v} \\ \frac{v-c_y}{f_y}\frac{\partial Z_C}{\partial u} & 1 + \frac{v-c_y}{f_y}\frac{\partial Z_C}{\partial v} \\ \frac{\partial Z_C}{\partial u} & \frac{\partial Z_C}{\partial v} \end{pmatrix} = \begin{pmatrix} \frac{f_x}{Z_C} & 0 \\ 0 & \frac{f_y}{Z_C} \end{pmatrix} \tag{25}$$

Therefore, at point $(x, y)$ on the image plane, the sampling rate for a single-camera system is:

$$\hat{\nu}_{sampling} = |J| = \frac{f_x f_y}{d^2} \tag{26}$$

where $d = Z_C(x, y)$. The overall sampling rate for a Gaussian primitive $p_k$ is given by:

$$\hat{\nu}_k = \max\left(\{\mathbb{V}_i(p_k) \cdot |J_i|\}_{i=1}^N\right) \tag{27}$$

where $N$ represents the number of cameras and $\mathbb{V}_i$ denotes the visibility function.

## B.2  Spatial Frequency of a 2D Gaussian Primitive

Given a spatial geometry with an analytic mathematical expression, the spatial frequency can be computed through the Spatial Fourier Transform (SFT).

The three-dimensional Fourier transform of a 2D Gaussian basis element is given by:

$$\mathcal{G}(\mathbf{k}) = \int_{\mathbb{R}^3} \mathcal{G}(u, v)\delta(\mathbf{p} - (\mathbf{p}_k + s_u \mathbf{t}_u u + s_v \mathbf{t}_v v))e^{-i\mathbf{k}\cdot\mathbf{P}}d\mathbf{p} \tag{28}$$

Since the Gaussian primitive is confined to the tangent plane, the integral can be simplified to a two-dimensional parameter space:

$$\mathcal{G}(\mathbf{k}) = s_u s_v e^{-i\mathbf{k}\cdot\mathbf{p}_k} \int_{-\infty}^{\infty} \int_{-\infty}^{\infty} \exp\left(-\frac{u^2 + v^2}{2}\right) e^{-i(s_u \mathbf{t}_u \cdot \mathbf{k} \cdot u + s_v \mathbf{t}_v \cdot \mathbf{k} \cdot v)} du\, dv \tag{29}$$

Applying the two-dimensional Gaussian integral formula:

$$\int_{-\infty}^{\infty} \int_{-\infty}^{\infty} e^{-\frac{1}{2}(u^2 + v^2) - i(au + bv)} du\, dv = 2\pi e^{-\frac{a^2 + b^2}{2}} \tag{30}$$

where $a = s_u \mathbf{t}_u \cdot \mathbf{k}$ and $b = s_v \mathbf{t}_v \cdot \mathbf{k}$. Substituting these values yields:

$$\mathcal{G}(\mathbf{k}) = 2\pi s_u s_v e^{-i\mathbf{k}\cdot\mathbf{p}_k} \exp\left(-\frac{s_u^2(\mathbf{t}_u \cdot \mathbf{k})^2 + s_v^2(\mathbf{t}_v \cdot \mathbf{k})^2}{2}\right) \tag{31}$$

The spatial frequency spectrum of a 2D Gaussian surfel is therefore determined by:

$$|\hat{G}(\mathbf{k})| = 2\pi s_u s_v \exp\left(-\frac{s_u^2(\mathbf{k} \cdot \mathbf{t}_u)^2 + s_v^2(\mathbf{k} \cdot \mathbf{t}_v)^2}{2}\right) \tag{32}$$

We define the projection of the wave vector $\mathbf{k}$ onto the tangent vector $\mathbf{t}_u$ as the spatial frequency $\nu_u$ in that direction. Since the Gaussian function contains over 95% of its energy within $\pm 2$ standard deviations, when considering a Gaussian of two standard deviations as the effective surfel size, the spatial frequency in the direction of $\mathbf{t}_u$ can be determined by the following condition:

$$-\frac{s_u^2(\mathbf{k} \cdot \mathbf{t}_u)^2 + s_v^2(\mathbf{k} \cdot \mathbf{t}_v)^2}{2} = -2, \quad \mathbf{k} \cdot \mathbf{t}_v = 0 \tag{33}$$

Thus, we obtain $\mathbf{t}_u \cdot \mathbf{k} = \frac{2}{s_u}$. Consequently, in the direction of $\mathbf{t}_u$, the angular frequency is $\omega_u = \mathbf{t}_u \cdot \mathbf{k} = \frac{2}{s_u}$ (and analogously, $\omega_v = \frac{2}{s_v}$ for the $\mathbf{t}_v$ direction).

Accounting for the $2\pi$ normalization convention of the Fourier transform, the spatial frequency of the Gaussian primitive along each tangent vector can be expressed as:

$$\nu_u = \frac{1}{\pi s_u}, \quad \nu_v = \frac{1}{\pi s_v} \tag{34}$$

# C  More Implementation Details and Experiments

## C.1  Network Design

In the initial feature extraction network $\Phi_{\text{image}}$, we implement a cross-view epipolar transformer and DINO feature backbones to extract preliminary image features $\mathcal{F}$. Subsequently, we employ the depth prediction network $\Phi_{\text{depth}}$ to regress per-view depth maps from the above image features. For the Gaussian feature prediction head $\Phi_{\text{attr}}$, we utilize a 2D convolutional network. The feature refinement network $\Phi_{\text{refine}}$ is implemented via a cross-attention network as described in the original paper.

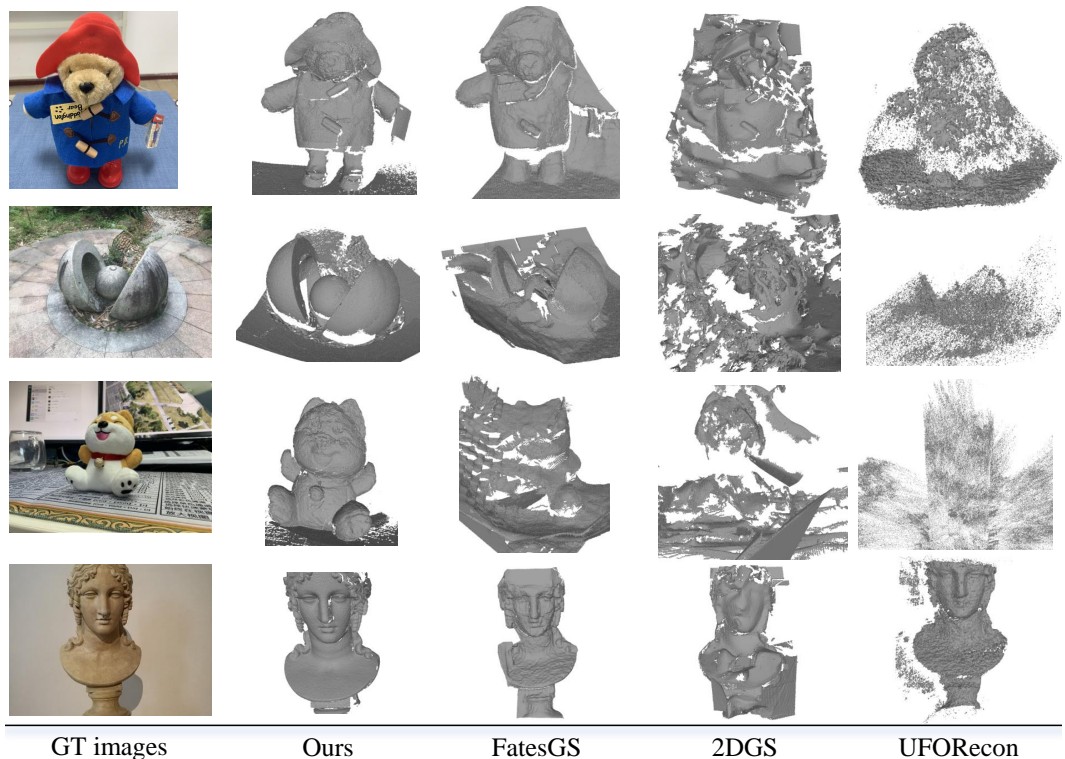

| GT images | Ours | FatesGS | 2DGS | UFORecon |

Figure 7: Visual comparison of 2-view reconstruction on BlendedMVS dataset.

## C.2 Baselines

We compare our method with SOTA methods with 2 categories. **i.** Neural implicit methods: NeuS [3], VolRecon [51], UFORecon [8], NeuSurf [6]. **ii.** Neural Explicit methods: 2DGS [18], GausSurf [19], FatesGS [64].

## C.3 Training Strategy

To progressively extract Gaussian features, we implement a two-phase curriculum learning-based training framework. During the initial phase, we leverage diverse scene datasets such as RealEstate10k [62]. Subsequently, in the refinement phase, we fine-tune the model on test sets from datasets such as DTU [61] that contain ground truth depth and surface measurements, thereby enhancing the precision of depth estimation and the characterization of geometric details.

## C.4 Experiments on BlendedMVS

We conduct experiments on the BlendedMVS dataset [63] and visualize the qualitative results in Figure 7. Given a pair of images, our method exhibits consistent and stable performance across all tested scenes after fine-tuning. In contrast, methods such as UFORecon [8] cannot maintain consistent performance across different scenes and may produce significant geometric collapse in certain scenarios. FatesGS [64] and 2DGS [18] achieve stable performance, but they tend to suffer from insufficient geometric consistency and fail to converge to a complete and smooth surface.

## C.5 Experiments on Novel View Synthesis

As shown in Figure 8, we further evaluate our approach through novel view synthesis experiments on the DTU dataset. Given a pair of input images, we synthesize intermediate viewpoints between the provided views and assess the quality of the generated novel views. We compare our method's visual fidelity against pixelSplat [23] and MVSplat [24]. To ensure a fair comparison, we fine-tune all baseline methods on the DTU training dataset and evaluate their performance on the designated

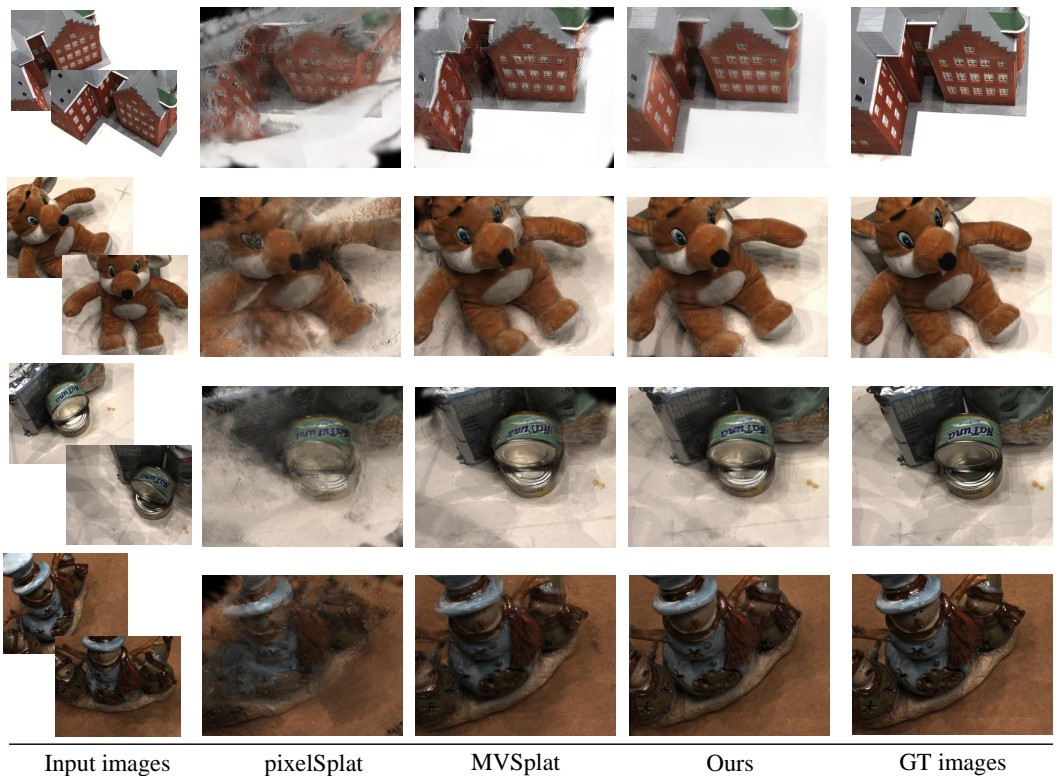

| Input images | pixelSplat | MVSplat | Ours | GT images |

Figure 8: Visual comparison of novel view synthesis on DTU dataset.

test scenes. As demonstrated in Figure 8, our method achieves superior novel view synthesis quality compared to existing approaches. This improvement can be attributed to our method's ability to capture fine-grained geometric details that are not preserved by alternative techniques.

### C.6 More Experimental Details

We constrain the Gaussian primitives to be either fully transparent or fully opaque, rather than semi-transparent. Consequently, the opacity attributes of 2D Gaussian Surfels are set to values approaching either 1 or 0 to facilitate clean surface extraction. The time consumption statistics reported in this paper represent the average inference time. The meshing process requires additional computational resources. On our hardware configuration, the meshing process consumes approximately 30 seconds.

## D Broader Impacts

The proposed Gaussian feed-forward network approach for fast surface reconstruction carries several notable societal implications. First, the acceleration of reconstruction processes may democratize access to high-quality 3D modeling capabilities across resource-constrained environments, potentially reducing technological disparities between well-funded research institutions and those with limited computational infrastructure. We also recognize the environmental impact dimension. While our method reduces computational requirements per reconstruction task, the aggregate environmental effect depends on whether this efficiency leads to reduced energy consumption or, conversely, to increased utilization through rebound effects. Future work should quantify these energy consumption patterns to better understand the net environmental impact.

## E Limitations and Future Works

**Camera Pose Configuration.** It is challenging for our approach to predict credible depth when input views only have small overlap regions. Our training data is relatively limited. We pretrain our

method on Re10K dataset (about 10,000), and subsequently perform fine-tuning on the DTU dataset. Methods such as VGGT [66] and Dust3R [67] demonstrate robust depth prediction capabilities across a wide range of camera configurations, benefiting from extensive training data (more than 1,000,000) with explicit depth regularization. The generalizability of our method is not enough.

**Efficiency.** The cost-volume techique predicts depth by computing correspondence between pairs of images, which indicates that processing images requires computational operations. Additionally, our methodology directly combines Gaussian groups derived from different viewpoints to construct scene representations, resulting in redundant representations particularly in overlapping regions where similar Gaussian primitives are predicted from multiple source images. Besides, the pixel-aligned Gaussians are sensitive to the resolution of input images. For high resolution inputs, e.g. $1024 \times 1024$, we generate over 1 million Gaussians for each view, which will significantly increase the inference and rendering time. The consumption of computational resources and time grows rapidly with more views or higher resolutions.

