# OpenReview forum: "SurfelSplat: Learning Efficient and Generalizable Gaussian Surfel Representations for Sparse-View Surface Reconstruction"
_NeurIPS.cc/2025/Conference — NeurIPS 2025 poster_

### Official Review · Reviewer_v9Gt · 2025-06-26

**Clarity:** 3
**Significance:** 3
**Originality:** 4
**Rating:** 5
**Confidence:** 5

**Summary:**

This paper presents a feed-forward (generalizable) framework for sparse-view surface reconstruction using Gaussian Splatting. The primary objective is to address the limitations of conventional feed-forward networks, which often misalign the covariance of Gaussians by orienting them parallel to the image plane. The authors analyze this issue from a signal processing perspective and attribute it to a violation of the Nyquist sampling theorem: the high spatial frequencies of pixel-aligned primitives exceed the system’s sampling rate, resulting in aliasing artifacts. To address this, the paper introduces two key contributions: (1) a Surfel Adaptation Module (Sec. 3.2.1), and (2) a Feature Aggregation Module (Sec. 3.2.2), both designed with guidance from the Nyquist theorem. Experiments on the standard DTU benchmark demonstrate that the proposed method achieves state-of-the-art performance in terms of Chamfer distance and computational efficiency.

**Questions:**

Based on the weaknesses discussed above, I would like to see the following points addressed during the rebuttal period:

1. Baseline Comparison: A more thorough discussion of the presented baselines is necessary, particularly regarding FateGS, which demonstrated state-of-the-art performance prior to this work. In addition, the paper appears to lack comparisons with other relevant baselines. For instance, recent feed-forward multi-view stereo methods such as Dust3R [1] or diffusion-based generalizable surface reconstruction MeshLRM [2], could provide meaningful points of reference and should be considered or at least discussed.

2. Justification of curriculum learning on Re10K: While the ablation study clearly demonstrates the effectiveness of the proposed modules, the role of pretraining on the Re10K dataset remains unclear. How crucial is this pretraining for achieving strong performance? Since some baselines may simply rely on more conventional backbones without such pretraining, a discussion or additional experiment aligned with those settings would strengthen the work.

3. Sparse View Sampling Strategy: More details are needed on how sparse views are sampled for both training and testing. In feed-forward approaches, generalization should be evaluated not only over scene distributions but also over diverse camera configurations. Clarifying this aspect would help readers understand the difficulty of the task ideally in the main paper, or at least in the appendix.

---

The points raised above are my primary concerns. Depending on how they are addressed during the rebuttal and reviewer discussion period, I may keep (accept) or lower (borderline accept to borderline reject) my final rating accordingly.

**Ethical Concerns:**

["NO or VERY MINOR ethics concerns only"]

**Final Justification:**

The initial recommendation of "accept" stands. The authors' rebuttal was clear and addressed all my questions with followup experimental validation. The paper's main contributions are its investigation of a key issue in generalizable surface reconstruction using 3DGS, and its state-of-the-art performance on sparse-view surface reconstruction. I recommend accepting the paper.

**Limitations:**

yes.

**Paper Formatting Concerns:**

I don't find any major formatting issues in this paper.

**Quality:**

4

**Strengths And Weaknesses:**

### Strengths

1.  The primary strength of this work lies in its novel analysis of why feed-forward networks struggle with sparse-view surface reconstruction using explicit primitives like Gaussian surfels. The authors identify the core issue as a fundamental violation of the Nyquist sampling theorem which could be occurred in pixel-aligned feed-forward setting. Framing the geometric inaccuracy—specifically, the tendency for surfels to overfit to the image plane—as an aliasing problem caused by the spatial frequency of primitives exceeding the sampling rate is a clever and original contribution in this field.

2. The paper's claims are backed by a solid theoretical framework; They provide a clear and valid mathematical formulation to quantify both the spatial sampling rate of a pixel in the multi-camera system and the spatial frequency of Gaussian surfels.

3. The proposed method is well-constructed, with each component logically addressing the problems identified in the initial analysis. The two main modules, the Nyquist-guided surfel adaptation and the cross-view feature aggregation, work in tandem to form a very coherent solution.

4. The paper demonstrates significant and meaningful improvements over a strong set of baselines. The experimental evaluation is conducted on the standard and challenging DTU benchmark. The proposed approach not only achieves state-of-the-art results in reconstruction accuracy, but it also achieves this with efficiency—offering inference in one second.

### Weaknesses
This is a solid paper with well-presented technical details, though several weaknesses remain that, if addressed, could further improve the work:

1. The paper could be strengthened with a more in-depth methodological comparison to key baselines. For instance, FatesGS is the top-performing baseline in Table 1, yet the paper lacks a discussion on the fundamental differences between it and the proposed method.

2. The necessity of the two-stage training strategy (pre-training on RealEstate10K before fine-tuning on DTU) is not fully justified. It is unclear how much of the performance gain is attributable to this extensive pre-training versus the novel modules themselves. A simple ablation study training directly on DTU from a standard backbones (e.g., DINO) would help isolate and more clearly validate the impact of the proposed components.

3. A more detailed explanation of how sparse views are sampled would be helpful. While the supplementary materials provide information on which input pairs are used during inference, this detail is not included in the main paper. Readers may be interested in how the input views are configured to better understand the difficulty of performing surface reconstruction from such limited observations in a feed-forward manner.

---

> ### Author Rebuttal · Authors · 2025-07-30
>
> ## 1. Baseline Comparisons
>
> ### 1.1 More thorough discussion of baselines, particularly FatesGS
> Thank you for the suggestion! FatesGS [1] deserves additional discussion as the second-best performer in reconstruction evaluations. FatesGS utilizes monocular depth ranking information to supervise depth distribution consistency within image patches, achieving fast runtime performance (~10 minutes). However, as an optimization-based method, FatesGS requires per-scene training and demonstrates relatively poor generalizability compared to our approach (shown in Supplementary Figure 1).
> Our method achieves high-fidelity Gaussian radiance fields through a highly efficient and generalizable feed-forward network. Regarding mesh quality (Figure 4), FatesGS produces complete but noisy surfaces, while our method extracts smoother results.
>
> ### 1.2 More comparisons with other relevant baselines
> We have evaluated against additional feed-forward networks and the mentioned works. Table A presents DTU benchmark results under consistent evaluation protocols, confirming our state-of-the-art performance among feed-forward approaches:
>
> **Table A Comparisons with feed-forward networks on DTU benchmarks**
> | | mean CD |
> |---|---|
> |pixelSplat [2] |10.53 |
> |MVSplat [3]|4.89 |
> |HiSplat [4]| 4.63|
> |MeshLRM [5]| 8.12|
> |Dust3R [6]| 3.55|
> |Ours |1.12 |
> > MeshLRM [5] remains closed-source, but we evaluated their HuggingFace demo.
>
> ## 2. Justification of curriculum learning on Re10K
>
> ### 2.1 Ablation study of pre-training stage on Re10k
> Thank you for this insightful comment. We chose Re10K pre-training to align with standard practices among Gaussian feed-forward networks. Now we evaluate performance without this costly pre-training stage. Table B shows that training solely on DTU achieves comparable DTU benchmark performance. However, Table C demonstrates that larger-scale pre-training significantly improves generalizability on BlendedMVS, following experimental settings from [7].
>
> **Table B Ablation study on pre-training with Re10k in DTU benchmarks**
> | | Mean CD|
> |---|---|
> |Ours |1.12 |
> |Ours w/o Re10k training | 1.14|
>
> **Table C Ablation study on pre-training with Re10k in BlendedMVS dataset**
> | | Mean CD|
> |---|---|
> |Ours | 0.57 |
> |Ours w/o Re10k pre-training |0.64 |
>
> ### 2.2 Some baselines may simply rely on more conventional backbones without such pretraining
> Thank you for this innovative suggestion! We experimented with adding DINO features to our backbone by concatenating DINO features extracted from input image pairs with our initial features in Equation 8, keeping the remaining network architecture unchanged. Results are shown in Table D, which demonstrate that DINO feature enhance the generalizability of our framework.
>
> **Table D Evaluation of our network plus DINO feature on BlendedMVS dataset**
> | | Mean CD|
> |---|---|
> |Ours + DINO | 0.56|
> |Ours +DINO w/o Re10k pre-training |0.60 |
>
> ## 3. Sparse View Sampling Strategy
>
> ### 3.1 Details on how sparse views are sampled
> Our main paper adopts FatesGS's **"large-overlap" setting** – the **standard sparse-view configuration** in this field. This provides 3 input views per scene; we randomly select 2 neighbor views for evaluation.
>
> > Configuration note:
> > - "Little-overlap": Originates from PixelNeRF [8]
> > - "Large-overlap": Defined in SparseNeuS [9]
>
>
> ### 3.2 Evalation on diverse camera configurations
> We conducted detailed experiments on both FatesGS camera configurations (large-overlap and little-overlap). Table E shows our method achieves superior reconstruction quality in both settings.
>
> **Table E Evaluation results on DTU benchmarks with 2 major camera configurations**
> | Mean CD $\downarrow$| large-overlap | little-overlap |
> |---|---|---|
> |FatesGS [1]| 1.18|1.57 |
> |2DGS [10]| 3.02| 3.52|
> |NeuSurf [11]| 1.44| 1.90|
> |Ours | 1.12| 1.55|
>
> ## References:
> > [1] Huang H, Wu Y, Deng C, et al. FatesGS: Fast and accurate sparse-view surface reconstruction using gaussian splatting with depth-feature consistency[C]//Proceedings of the AAAI Conference on Artificial Intelligence. 2025, 39(4): 3644-3652.\
> > [2] Charatan D, Li S L, Tagliasacchi A, et al. pixelsplat: 3d gaussian splats from image pairs for scalable generalizable 3d reconstruction[C]//Proceedings of the IEEE/CVF conference on computer vision and pattern recognition. 2024: 19457-19467.\
> > [3] Chen Y, Xu H, Zheng C, et al. Mvsplat: Efficient 3d gaussian splatting from sparse multi-view images[C]//European Conference on Computer Vision. Cham: Springer Nature Switzerland, 2024: 370-386.\
> > [4] Tang S, Ye W, Ye P, et al. Hisplat: Hierarchical 3d gaussian splatting for generalizable sparse-view reconstruction[J]. arxiv preprint arxiv:2410.06245, 2024.\
> > [5] Wei X, Zhang K, Bi S, et al. Meshlrm: Large reconstruction model for high-quality meshes[J]. arxiv preprint arxiv:2404.12385, 2024.\
> > [6] Wang S, Leroy V, Cabon Y, et al. Dust3r: Geometric 3d vision made easy[C]//Proceedings of the IEEE/CVF Conference on Computer Vision and Pattern Recognition. 2024: 20697-20709.\
> > [7] Zhang Y, Hu Z, Wu H, et al. Towards unbiased volume rendering of neural implicit surfaces with geometry priors[C]//Proceedings of the IEEE/CVF Conference on Computer Vision and Pattern Recognition. 2023: 4359-4368.\
> > [8] Yu A, Ye V, Tancik M, et al. pixelnerf: Neural radiance fields from one or few images[C]//Proceedings of the IEEE/CVF conference on computer vision and pattern recognition. 2021: 4578-4587.\
> > [9] Long X, Lin C, Wang P, et al. Sparseneus: Fast generalizable neural surface reconstruction from sparse views[C]//European Conference on Computer Vision. Cham: Springer Nature Switzerland, 2022: 210-227.\
> > [10] Huang B, Yu Z, Chen A, et al. 2d gaussian splatting for geometrically accurate radiance fields[C]//ACM SIGGRAPH 2024 conference papers. 2024: 1-11.\
> > [11] Huang H, Wu Y, Zhou J, et al. NeuSurf: On-surface priors for neural surface reconstruction from sparse input views[C]//Proceedings of the AAAI conference on artificial intelligence. 2024, 38(3): 2312-2320.

---

> > ### Comment · Reviewer_v9Gt · 2025-08-04
> >
> > Thank you for the detailed response. All of my concerns have been clearly addressed.
> > While the authors mention the limitations and future directions in the supplementary materials, the discussion remains somewhat limited. I encourage the authors to elaborate further with a more detailed and insightful discussion of their limitations (e.g., assumptions on camera poses), potential failure cases (e.g., under what conditions the method might fail), and future research directions.

---

> > > ### Author Response · Authors · 2025-08-07
> > >
> > > Thank you for your constructive feedback. We appreciate your suggestion to provide a more comprehensive discussion of the limitations, failure cases and future research directions.
> > >
> > > ## 1. Limitations
> > >
> > > **1.1 Camera Pose Configuration.** It is challenging for our approach to predict credible depth when input views only have small overlap regions. On one hand, our training data is relatively limited. We pretrain our method on Re10K dataset (about 10,000), and subsequently perform fine-tuning on the DTU dataset.
> > > Methods such as VGGT [1] and Dust3R [2] demonstrate robust depth prediction capabilities across a wide range of camera configurations, benefiting from extensive training data (more than 1,000,000) with explicit depth regularization.
> > > On the other hand, the cost-volume-based depth estimator requires sufficient visual correspondence between views to function properly [3].
> > > Thus, their accuracy deteriorates substantially as the geometric overlap between views decreases. When the baseline between cameras becomes too large, the shared visual content becomes insufficient for establishing reliable correspondences. Consequently, the depth estimation process fails to produce accurate geometric information, leading to corrupted reconstruction results.
> > >
> > > **1.2 Efficiency.** The cost-volume techique predicts depth by computing correspondence between pairs of images, which indicates that processing $N$ images requires $O(N^2)$ computational operations. Additionally, our methodology directly combines Gaussian groups derived from different viewpoints to construct scene representations, resulting in redundant representations particularly in overlapping regions where similar Gaussian primitives are predicted from multiple source images.
> > > Besides, the pixel-aligned Gaussians are sensitive to the resolution of input images. For high resolution inputs, e.g. 1024 × 1024, we generate over 1 million Gaussians for each view, which will significantly increase the inference and rendering time.
> > > The consumption of computational resources and time grows rapidly with more views or higher resolutions.
> > >
> > > ## 2. Failure Cases
> > > The aforementioned limitations manifest most prominently when processing widely-separated viewpoints. Under these conditions, the depth estimation module produces unreliable predictions, potentially leading to reconstruction failures.
> > >
> > > ## 3. Future Works
> > > To address these fundamental limitations, our future research will pursue several complementary directions:
> > >
> > > **3.1 Robustness on Camera Poses:** We will further explore the scaling up strategy on training data and model parameters with a simpler architecture, such as incorporating datasets with comprehensive depth annotations, including BlendedMVS [4] and MegaDepth [5], or adopting state-of-the-art depth estimation architectures such as MonSter [6] and LightStereo [7]. Besides, the advancement in 3D foundation model significantly promotes the surface reconstruction field, which will enhance the robustness in practice.
> > >
> > > **3.2 Efficiency:** Our next step aims to real-time surface reconstruction.
> > > Inspired by Fast3R [8], which employs Transformer-based feature fusion modules to enable bi-directional information flow and simultaneous dense input processing, we aim to establish a unified framework capable of recovering scene geometry from extended image sequences while maintaining both robustness and real-time performance.
> > > Also, we intend to model the relations of Gaussians to reduce redundancy in overlap regions and low-frequency areas and achieve more efficient scene representations.
> > >
> > > ## References:
> > > > [1] Wang J, Chen M, Karaev N, et al. Vggt: Visual geometry grounded transformer[C]//Proceedings of the Computer Vision and Pattern Recognition Conference. 2025: 5294-5306.\
> > > > [2] Wang S, Leroy V, Cabon Y, et al. Dust3r: Geometric 3d vision made easy[C]//Proceedings of the IEEE/CVF Conference on Computer Vision and Pattern Recognition. 2024: 20697-20709.\
> > > > [3] Caliskan A, Mustafa A, Imre E, et al. Learning dense wide baseline stereo matching for people[C]//Proceedings of the IEEE/CVF International Conference on Computer Vision Workshops. 2019: 0-0.\
> > > > [4] Yao Y, Luo Z, Li S, et al. Blendedmvs: A large-scale dataset for generalized multi-view stereo networks[C]//Proceedings of the IEEE/CVF conference on computer vision and pattern recognition. 2020: 1790-1799.\
> > > > [5] Li Z, Snavely N. Megadepth: Learning single-view depth prediction from internet photos[C]//Proceedings of the IEEE conference on computer vision and pattern recognition. 2018: 2041-2050.\
> > > > [6] Cheng J, Liu L, Xu G, et al. Monster: Marry monodepth to stereo unleashes power[C]//Proceedings of the Computer Vision and Pattern Recognition Conference. 2025: 6273-6282.\
> > > > [7] Guo X, Zhang C, Zhang Y, et al. LightStereo: Channel Boost Is All You Need for Efficient 2D Cost Aggregation[J], 2024.\
> > > > [8] Yang J, Sax A, Liang K J, et al. Fast3r: Towards 3d reconstruction of 1000+ images in one forward pass[C]

---

### Official Review · Reviewer_D7r6 · 2025-06-27

**Clarity:** 2
**Significance:** 3
**Originality:** 3
**Rating:** 4
**Confidence:** 3

**Summary:**

This paper proposes a generalizable 3DGS method for sparse view surface reconstruction. It focuses on dealing with the low frequency issue when projecting Gaussian primitives to input view grids due to their relatively small scales. After the commonly used generalizable 3DGS prediction, they compute the maximum spatial frequency of each gaussian, and use a low-pass filter to adjust their scales. After this, they aggregate the per-view features by projecting gaussians to each view, and re-predict the gaussian parameters.

**Questions:**

Please see the questions in the weaknesses above.

**Ethical Concerns:**

["NO or VERY MINOR ethics concerns only"]

**Limitations:**

See the weaknesses.

**Paper Formatting Concerns:**

N.A.

**Quality:**

3

**Strengths And Weaknesses:**

Strengths:
1. Significance: The proposed method focuses on a novel viewpoint of gaussian surface reconstruction based on Nyquist Conditions.
2. Results: The experiment results evaluate the state-of-the-art performance of the proposed method.

Weaknesses:
1. Clarity: I find some parts of this paper to be unclear:

 (i) The gaussian adaptation to increase their scales are performed before the feature aggregation step, however the final gaussian parameters are re-predicted by the same output head as the initial head. Then how to make sure that the final gaussians $\hat{f}_i$ can satisfy the Nyquist Conditions? Or the Nyquist Conditions is just to improve the feature aggregation step?

 (ii) Is there any constraints on the initial gaussian attributes $f_i$? Are the depth and normal losses also conducted on the initial gaussian attributes? If not then the Figure 5 comparison is unfair.

---

> ### Author Rebuttal · Authors · 2025-07-30
>
> ### 1. How to make sure that the final gaussians $\hat f_i$ can satisfy the Nyquist Conditions?
> Our method involves three stages of Gaussian primitives: initial Gaussians $f_i$, adapted Gaussians $f_i^{adapted}$ and the final Gaussians $\hat f_i$. Only the adapted Gaussians $f_i^{adapted}$ need to satisfy the Nyquist theorem.
> The surfel adaptation module's primary purpose is to gather sufficient information for accurate geometric attribute prediction. Its key function is obtaining image regions $\mathcal R_i$ associated with Gaussian surfel $f_i$ regression (descibed in line 176). Once the adapted Gaussian attributes satisfy Nyquist conditions, we can recover the real signal $\hat f_i$ from features in image regions $\mathcal R_i$.
>
> Consequently, final Gaussians $\hat f_i$ need not satisfy Nyquist conditions. Our goal is achieving surfels with correct spatial geometric orientation, which the feature aggregation module guarantees. Forcing final Gaussians to satisfy Nyquist conditions would expand surfels approximately 10-fold, compromising fine geometric detail representation.
>
> To demonstrate this, we evaluated reconstruction on DTU benchmarks with an additional surfel adaptation module after $\hat f_i$. As shown in Table A, the performance of our method decreased significantly.
>
> **Table A Evaluation on additional surfel adaption module**
> | | Mean CD |
> |---|---|
> | Ours | 1.12 |
> | Ours plus another surfel adaption module | 4.32 |
>
> ### 2. Is there any constraints on the initial gaussian attributes $f_i$? Are the depth and normal losses also conducted on the initial gaussian attributes?
>
> For the initial Gaussian attributes $ f_i $, we applied identical geometric constraints (depth and normal losses) during training. Despite these constraints, normal maps were incorrectly recovered. This demonstrates that satisfying the sampling theorem is essential for learning correct geometric distributions, regardless of geometric constraints.
>
> Figure 5 presents initial Gaussian attributes without geometric constraints for more striking comparison. For fairness, we conducted additional DTU experiments with consistent settings:
>
> **Table B Reconstruction results with Geometric losses**
> | |Normal Loss | Normal Covariance| Mean CD |
> |---|---| ---|---|
> | MVSplat | 0.148 | 0.023|8.33|
> | MVSplat w/ $Loss_{geo}$ | 0.140 |0.025 | 4.89|
> | Ours | 0.060 |0.540 | 1.12|
>
> In the revised manuscript, we will add this in Section 4.2 for fair comparisons. Thank you for your constructive feedback, which has significantly improved our paper!

---

> > ### Comment · Area_Chair_nXww · 2025-08-03
> >
> > Reviewer D7r6, did the rebuttal address your concerns? Do you have any further questions or comments for the authors?

---

> > > ### Comment · Reviewer_D7r6 · 2025-08-07
> > >
> > > Thank you for your reponse, which has solved my questions. I will keep my rating.

---

### Official Review · Reviewer_pJpP · 2025-06-30

**Clarity:** 1
**Significance:** 2
**Originality:** 2
**Rating:** 2
**Confidence:** 5

**Summary:**

The paper investigates the problem of 3D scene surface reconstruction using Gaussian Splatting techniques. Specifically, the SurfaceSplat method is proposed as a feed-forward framework to generate efficient and generalizable surfel representations from sparse-view images. The main observation of the paper is that the spatial frequency of pixel-aligned Gaussian surfels exceeds the Nyquist sampling rate in existing works, which inspires the authors to modulate the geometric forms of diverse Gaussian surfels in the frequency domain and correlate pixel regions across multiple input views towards better Gaussian geometric feature learning. Experiments on the DTU dataset demonstrate the method's advantage in sparse-view surface reconstruction.

**Questions:**

Please respond to the points made in the Weaknesses section above.

**Ethical Concerns:**

["NO or VERY MINOR ethics concerns only"]

**Final Justification:**

After reading the rebuttals and other reviews, I still find the paper limited by both experimental validation and theoretical justification.

Experiment-wise, the results in the original paper are not convincing, as the proposed method uses more data and a different training setting than previous methods. The new results in the rebuttal mitigated the issue, but it is still unclear if all methods are trainied with RealEstate + DTU datasets.

More importantly, the theoretical justification is very weak. The paper and the rebuttal did not convince me regarding how Nyquist sampling theorem is applied. The "real signal" is the true surface, while the authors use the 2D Gaussian primitives as its proxy, which makes little sense, as there is no guarantee that the Gaussians approximate the surface well. In fact, they usually do not. Furthermore, I did not understand how frequency of Gaussian surfel is related to the maximum frequency of the real signal that Nyquist sampling theorem requires.

For the above reasons, I keep my recommendation as rejection.

**Limitations:**

Yes

**Quality:**

2

**Strengths And Weaknesses:**

## Strengths

The paper observes an interesting phenomenon in applying 2D Gaussian Splatting to learned surface reconstruction and provides a theoretical explanation.

The solution is simple and intuitive.

## Weaknesses

The presentation of the paper is confusing. Too much important information is deferred to supplementary material, rendering the main text hard to follow. I suggest Section 1.1 from supp. be merged with the main text; otherwise, the paper lacks explanations for many basic concepts and terms.

Regarding the main contribution, i.e., the adaptation based on the Nyquist sampling theorem, I find it flawed in the following two aspects;
i) The authors are using the overall sampling frequency as the maximum frequency across all views, which lacks theoretical justification. Sampling is inherently additive with more views, and occlusion is not taken into consideration in this approximation.
ii) I did not understand what the "real signal" to recover is. Based on Eq.1, I assumed the real signal should be the ground truth surface as a 2D signal in a given view. However, the authors are applying Nyquist sampling theorem with spatial frequency of 2D Gaussian primitives, what is the loginc of this? How is that the "real signal" that we are supposed to sample from?

The experiments are not convincing. For the compared methods, are they re-trained for fairness, i.e., on the same data for two-view reconstruction?

---

> ### Author Rebuttal · Authors · 2025-07-31
>
> ### 1. Too much important information is deferred to supplementary material
> We apologize for placing substantial content in the supplementary materials due to the NeurIPS page limitations. In the final version, we will integrate key derivations from the supplementary materials into the main text.
> Here are the specific improvements we will merge Section 1.1 from the supplementary material into Section 3.1 in the main text, as you suggested. This section contains crucial explanations of:
> * Basic definitions of spatial frequency and sampling rates from line 43 to line 77;
> * The relationship between pixel-aligned primitives and Nyquist constraints from line 79 to line 83.
>
> If you have any question and recommendation, we welcome discussion during the discussion period.
>
> ### 2. The use of maximum sampling frequency:
>
> We sincerely appreciate the reviewer's insightful critique regarding our Nyquist-based adaptation.
> Our choice of using the maximum sampling frequency $\hat \nu_k = max(({\mathbf V_i(p_k) \cdot |J_i|})^N_{i=1})$ (Equation 3) is motivated by the design of Mip-Splatting [1] (Equation 7).
>
> The key insight of Mip-Splatting is that for accurate reconstruction, we need to ensure that each 3D Gaussian primitive satisfies the Nyquist sampling criterion for **at least one camera view** where it is visible. This is because if a primitive can be accurately reconstructed from at least one view, we have captured its essential geometric information. Using max ensures we respect the highest sampling rate available, preventing aliasing in the view with the finest sampling.
> By selecting the maximum frequency as our sampling frequency, our surfel adaptation module guarantees that the network captures **sufficient** geometric information from at least one input view, which is **sufficient** for recovering Gaussian surfels with fine geometric details.
>
> **Alternative design choices and their limitations:**
> We considered two alternative additive aggregation strategies: Sum of **sampling frequencies** and **Root mean square of sampling frequencies**.
> They would overestimate the effective sampling frequency, making it unlikely to satisfy the Nyquist criterion from any single perspective.
> To validate our design choice, we conducted experiments with different frequency aggregation strategies. As shown in Table A, we evaluate surface reconstruction results on DTU benchmarks, and the result shows that our method can present better reconstruction quality.
>
> **Table A  Experiments on different final frequency settings**
> | | Mean CD $\downarrow$|
> |---|---|
> |$ \sum_i^N \nu_i $ | 1.28|
> |$ \sqrt{\sum_i^N \nu_i^2}  $ | 1.25|
> |$max(\nu_i)$ | 1.12 |
>
>
> We will enhance our manuscript by:
>
> (1) Acknowledging Mip-Splatting [1] as the motivation for our maximum frequency selection;
>
> (2) Providing detailed theoretical justification in Section 3.1.2 explaining why the maximum frequency is both sufficient and theoretically sound for accurate surfel prediction.
>
>
>
> ### 3. Occlusion is not taken into consideration in this approximation
> We appreciate the reviewer's attention to occlusion handling. In our current implementation, the visibility function $V_i(p_k)$ in Equation 3 accounts for basic visibility by checking if the Gaussian center falls within the view frustum. Similar to Mip-Splatting [1], we found the current approach sufficient for our experiments.
>
>
> ### 4. Clarification on the "real signal" to recover
>
> **What is the "real signal" we want to recover?**
>
> The real signal we aim to recover is the 3D surface geometry $\mathcal M$ of the scene. However, we only have access to discrete 2D image observations of this continuous 3D signal.
> And 2D Gaussian surfels are our chosen representation to approximate this surface.
>
> Specifically, we model the real signal $\mathcal M$ using a collection of 2D Gaussian primitives (following the foundation established by 2DGS [2] and Gaussian Surfels [3]). The problem of reconstructing the surface from discrete 2D sampling is thus reformulated as reconstructing the Gaussian primitives from the 2D image data.
>
> **Why apply Nyquist theorem to 2D Gaussian primitives?**
>
> To reconstruct Gaussian primitives accurately, we need to analyze three key components:
> * Spatial frequency of representation elements: Each 2D Gaussian surfel has an inherent spatial frequency $ \nu$(Equation 4).
> * Spatial sampling frequency from 2D image: Given the sampling density in the image plane, we can compute the sampling frequency $\hat \nu$ (Equation 2)
> * Nyquist Sampling theorem constraint: To accurately represent a signal component with frequency $\nu$, we need sampling rate $\hat \nu \geq 2\nu$.
>
> By applying the Nyquist theorem to 2D Gaussian primitives, we can identify which Gaussian elements violate the Nyquist sampling criterion. Our network design then specifically addresses these under-sampled primitives to ensure complete recovery of all Gaussian elements, thereby achieving a high-fidelity approximation of the 3D surface geometry.
>
> ### 5. Experimental fairness concerns
> Thank you for raising this important point about experimental fairness. Let us clarify our experimental setup and provide additional results to address your concern.
>
> **Experimental setup clarification:** For the methods compared in our experiments:
>
> * Methods that don't require task-specific training: Some approaches like 2DGS [2] and FatesGS [4] are optimization-based methods that directly optimize on the given input views without requiring pre-training. These methods were run directly on the 2-view inputs.
> * Methods with pre-trained models: For learning-based methods (VolRecon [5], UFORecon [6]), we acknowledge that we initially used their publicly available pre-trained models, which were trained on 3-view datasets. We recognize this could introduce bias in the comparison.
>
> **Additional experiments with re-trained models:**
> To address this fairness concern, we have conducted additional experiments where we re-trained the top-performing learning-based methods specifically for 2-view reconstruction. As shown in Table B, even with task-specific training, our method maintains superior performance while offering higher efficiency. We will include these results and clarify the experimental setup in the revised manuscript to ensure transparency.
>
> **Table B Additional 2-view training on VolRecon and UFORecon.**
> |ID|24|37|40|55|63|65|69|83|97|105|106|110|114|118|122|Mean|
> |---|---|---|---|---|---|---|---|---|---|---|---|---|---|---|---|---|
> |VolRecon [5]|1.32|3.03|1.66|1.42|1.64|2.11|1.40|1.74|1.49|1.25|1.50|1.52|0.95|1.34|1.63| 1.60 |
> |UFORecon [6]|1.12|2.22|1.61|2.53|1.72|2.40|1.46|1.40|2.02|0.93|2.10|1.87|1.34|1.98|1.55| 1.75|
> |Ours|1.23|2.64|1.63|0.90|1.24|1.14|1.12|1.18|1.13|0.79|0.84|0.54|0.51|0.84|1.04|1.12|
>
> ## References:
> > [1] Yu Z, Chen A, Huang B, et al. Mip-splatting: Alias-free 3d gaussian splatting[C]//Proceedings of the IEEE/CVF conference on computer vision and pattern recognition. 2024: 19447-19456.\
> > [2] Huang B, Yu Z, Chen A, et al. 2d gaussian splatting for geometrically accurate radiance fields[C]//ACM SIGGRAPH 2024 conference papers. 2024: 1-11.\
> > [3] Dai P, Xu J, **e W, et al. High-quality surface reconstruction using gaussian surfels[C]//ACM SIGGRAPH 2024 conference papers. 2024: 1-11.\
> > [4] Huang H, Wu Y, Deng C, et al. FatesGS: Fast and accurate sparse-view surface reconstruction using gaussian splatting with depth-feature consistency[C]//Proceedings of the AAAI Conference on Artificial Intelligence. 2025, 39(4): 3644-3652.\
> > [5] Ren Y, Wang F, Zhang T, et al. Volrecon: Volume rendering of signed ray distance functions for generalizable multi-view reconstruction[C]//Proceedings of the IEEE/CVF Conference on Computer Vision and Pattern Recognition. 2023: 16685-16695.\
> > [6] Na Y, Kim W J, Han K B, et al. Uforecon: Generalizable sparse-view surface reconstruction from arbitrary and unfavorable sets[C]//Proceedings of the IEEE/CVF Conference on Computer Vision and Pattern Recognition. 2024: 5094-5104.

---

> > ### Comment · Area_Chair_nXww · 2025-08-03
> >
> > Reviewer pJpP, did the rebuttal address your concerns? Do you have any further questions or comments for the authors?

---

> > ### Comment · Reviewer_pJpP · 2025-08-06
> >
> > Dear all,
> >
> > After reading the rebuttal and other reviews, I am still inclined to keep my original rating to reject the paper.
> >
> > I appreciate the authors' efforts to provide new experimental results of retrained models. However, it really should have been done before submitting the paper. And the new results did not fully address my concerns. Testing on the 2-view setting differs from the setting in the compared methods and makes the results less convincing.
> >
> > More importantly,  the rebuttal did not convince me regarding the justification of how Nyquist sampling theorem is applied. The "real signal" is the true surface, while the authors use the 2D Gaussian primitives as its proxy, which makes little sense, as there is no guarantee that the Gaussians approximate the surface well. In fact, they usually do not. Furthermore, I did not understand how frequency of Gaussian surfel is related to the maximum frequency of the real signal that Nyquist sampling theorem requires. Overall, I think the theoretical justification is flawed.

---

> > > ### Author Response · Authors · 2025-08-08
> > > **Regarding Experimental Settings**
> > >
> > > We appreciate your concern about experimental settings. We maintain that our evaluation approach provides meaningful insights for the following reasons:
> > >
> > > **Primary Evaluation Rationale:** After successfully introducing feed-forward networks to surface reconstruction, our method's core advantages lie in its strong generalization capability and ability to leverage pre-trained priors. To test these advantages, we specifically adopted the 2-view setting as our primary benchmark (Randomly choose 2 neighbor views in "little-overlapping" setting [1]). This extremely sparse-view configuration best highlights our method's strengths—while optimization-based methods struggle with insufficient view constraints in such limited settings, our feed-forward approach can effectively utilize learned priors to compensate for the lack of input information, demonstrating superior robustness and generalization.
> > >
> > > **Comprehensive Validation:** To provide a complete evaluation, we also conducted experiments using the standard 3-view "little-overlapping" setting from PixelNeRF [1]. As shown in Table R1, with increased input views, optimization-based methods naturally benefit from richer information, partially attenuating our performance gap. Despite this, our approach still achieves competitive reconstruction quality (Mean CD: 1.39) while maintaining overwhelming efficiency advantages—operating 50,000× faster than NeuSurf [2] (1 second vs. 14 hours) and 600× faster than FatesGS [3] (1 second vs. 10 minutes).
> > >
> > > **Table R1 Reconstruction performance on little-overlapping settings**
> > >
> > > |ID|24|37|40|55|63|65|69|83|97|105|106|110|114|118|122|Mean CD| Inference Time|
> > > |---|---|---|---|---|---|---|---|---|---|---|---|---|---|---|---|---|---|
> > > |VolRecon [4]|3.05 |4.45| 3.36 |3.09| 2.78| 3.68 |3.01 |2.87 |3.07 |2.55 |3.07 |2.77 |1.59 |3.44| 2.51 |3.02|30 seconds |
> > > |UFORecon [5]|1.52| 2.58| 1.85| 1.44| 1.55| 1.81| 1.06| 1.52 |0.96 |1.40| 1.19 |0.94| 0.65| 1.25 |1.29| 1.40|60 seconds |
> > > |NeuS [6]|4.11 |5.40 |5.10 |3.47 |2.68 |2.01 |4.52| 8.59| 5.09 |9.42 |2.20 |4.84 |0.49| 2.04 |4.20 |4.28| 10 hours|
> > > |NeuSurf [2]|1.35 |3.25| 2.50 |0.80 |1.21 |2.35 |0.77| 1.19| 1.20| 1.05 |1.05| 1.21| 0.41| 0.80| 1.08| 1.35| 14 hours |
> > > |2DGS [7]|3.25|3.64|3.52| 1.42| 2.04 |2.52| 1.99| 2.69| 2.55| 1.79| 2.92| 4.50| 0.73 |2.38 |1.79| 2.52|14 mins |
> > > |FatesGS [3]| 1.32 |2.85| 2.71| 0.80 |1.44 |2.08 |1.11| 1.19| 1.33 |0.76| 1.49 |0.85 |0.47 |1.05| 1.06| 1.37| 10 mins|
> > > |Ours|1.54|2.85|1.93|1.22|1.61|1.23|1.32|1.56|1.45|0.98|1.05|0.62|0.86|1.11|1.52|1.39| 1 second |

---

> > > > ### Author Response · Authors · 2025-08-08
> > > > **Theoretical Analysis of Signal Sampling and Reconstruction with Gaussian Kernels**
> > > >
> > > > In signal processing, continuous signals cannot be stored or processed directly by digital systems, making the sampling of continuous signals and the reconstruction of the original signal from discrete samples critical. For a continuous time-domain signal with a maximum frequency $\nu_{max}$ in the frequency domain, the Nyquist-Shannon sampling theorem guarantees perfect reconstruction if the sampling frequency $\hat{\nu}_{sampling}$ satisfies:
> > > >
> > > > $\hat \nu_{sampling} \geq 2\nu_{max}$.
> > > >
> > > > This theorem is derived from the fact that sampling in the time domain is equivalent to periodic extension in the frequency domain, with the period equal to the sampling frequency $\hat{\nu}_{sampling}$.
> > > >
> > > > To prevent aliasing, where spectral components overlap and become indistinguishable, the period of extension must exceed twice the maximum signal frequency $\nu_{max}$, ensuring separability of the frequency-domain representations [8].
> > > >
> > > > To recover the frequency-domain information of the original signal $f(x)$, a low-pass filter $G(\omega)$ is applied to the Fourier transform $F(\omega)$ of the sampled signal to eliminate the effects of periodic extension introduced by sampling. Mathematically, multiplying the Fourier transform $F(\omega)$ by a low-pass filter $G(\omega)$ is equivalent to convolving the sampled signal $f(x)$ with the inverse Fourier transform $g(x)$ of the filter, which is known as the reconstruction kernel and defined at the sampling points. A Gaussian function is often chosen as the reconstruction kernel due to its Fourier transform being an effective low-pass filter with desirable smoothness properties [9].
> > > >
> > > > In the context of 3D Gaussian Splatting (3DGS), the objective is to reconstruct a three-dimensional color field, for which three-dimensional Gaussians are selected as the reconstruction kernel. For surface reconstruction, the goal is to reconstruct a surface field. However, the surface field exhibits infinite frequency content in the normal direction. As the Fourier transform of a Dirac delta function $\delta(x)$ (representing the normal direction) is a constant function, it contains all frequency components with non-selective frequency characteristics. Consequently, the 2D Gaussian function is chosen as the reconstruction kernel for surface fields to better accommodate these properties as used in 2DGS[7], FatesGS[3].
> > > >
> > > > In our surface reconstruction framework, the continuous signal to be recovered is the true surface, while the sampling signal is the input images composed of discrete pixels. Given fixed pixel size, camera focal length, and depth, the sampling frequency $\hat{\nu}_{sampling}$ of the discrete signal is predetermined. According to the Nyquist-Shannon sampling theorem, the maximum recoverable signal frequency is:
> > > >
> > > > $\nu_{max} \le \dfrac{\hat{\nu}_{sampling}}{2}$.
> > > >
> > > >
> > > > If the true surface contains frequency components exceeding $\frac{\hat \nu_{sampling}}{2} $,  perfect reconstruction is theoretically impossible due to aliasing. In practice, direct manipulation of the true surface is infeasible, and we approximate it using Gaussian kernels. However, the maximum recoverable frequency remains constrained by the sampling frequency, and high-frequency components beyond $\frac{\hat{\nu}_{sampling}}{2} $ are unattainable. If the Gaussian kernel includes too much high-frequency components (e.g. pixel-aligned Gaussians in MVSplat), they cannot be accurately recovered, leading to incorrect geometric features learned by the neural network.
> > > >
> > > > To address the theoretical unavailability of high-frequency information, we adopt a strategy of restricting the frequency of the Gaussian kernels. By constraining the Gaussians to lower frequencies, the geometric features of the reconstructed surface are more accurately represented within the limits of the sampling frequency, resulting in a more complete and reliable surface reconstruction.
> > > >
> > > > ## References
> > > > > [1] Yu A, Ye V, et al. pixelnerf: Neural radiance fields from one or few images\
> > > > > [2] Huang H, Wu Y, et al. NeuSurf: On-surface priors for neural surface reconstruction from sparse input views\
> > > > > [3] Huang H, Wu Y, et al. FatesGS: Fast and accurate sparse-view surface reconstruction using gaussian splatting with depth-feature consistency\
> > > > > [4] Ren Y, Wang F, et al. Volrecon: Volume rendering of signed ray distance functions for generalizable multi-view reconstruction\
> > > > > [5] Na Y, Kim W J, et al. Uforecon: Generalizable sparse-view surface reconstruction from arbitrary and unfavorable sets\
> > > > > [6] Wang P, Liu L, Liu Y, et al. Neus: Learning neural implicit surfaces by volume rendering for multi-view reconstruction\
> > > > > [7] Huang B, Yu Z, Chen A, et al. 2d gaussian splatting for geometrically accurate radiance fields\
> > > > > [8] C. E. Shannon, “Communication in the presence of noise” \
> > > > > [9] A. V. Oppenheim, R. W. Schafer, and J. R. Buck, Discrete-Time Signal Processing, 2nd ed.

---

### Official Review · Reviewer_Wk3j · 2025-06-30

**Clarity:** 1
**Significance:** 3
**Originality:** 3
**Rating:** 4
**Confidence:** 4

**Summary:**

The authors propose SurfaceSplat, a feed-forward approach for multi-view surface reconstruction using 2D Gaussian surfels that is fast and improves on the state-of-the-art. They observed a problem with conventional feed-forward approaches where the geometric attributes of Gaussian surfels were not accurately recovered. Specifically, the normal vectors often oriented parallel to the image plane instead of aligning with the actual surface, because their spatial frequency exceeded Nyquist sampling rates.

Thus, they propose a Nyquist theorem-guided Gaussian surfel adaptation module to constrain the maximum frequency of the Gaussian surfels to fall within Nyquist thresholds. The theoretical analysis for the surfel adaptation module is provided, demonstrating its ability to adjust frequencies, and its effectiveness is then experimentally verified. The results show that these Nyquist constraints significantly improve depth estimation and normal rendering performance in feed-forward networks.

**Questions:**

**Considering the weaknesses in motivation:**
- What are the main references for the networks that presented the issues mentioned in line 50?
- Can the authors provide a concrete example?

**Considering the weaknesses in efficiency:**
- Why is your method so fast? What improvements in your frequency modeling also improve speed?
- What is the difference in speed between your approach and pixelSplat (the backbone used)?

**Considering the weaknesses in the results:**
- If feed-forward networks are the motivation for the authors' solution, why are they not featured in the experiments?
- Could the proposed solution be used as an add-on to previous methods?

**Ethical Concerns:**

["NO or VERY MINOR ethics concerns only"]

**Final Justification:**

After careful consideration of the authors' rebuttal and the significant work they've put in, I am increasing my score to reflect my confidence in the paper. They have provided a thorough justification of their approach and have committed to several key changes for a clearer presentation. Specifically, their clarifications on the efficiency claims and backbones, along with new experimental results that fully validate their claims, have resolved my previous concerns. I am increasing the grade towards acceptance.

**Limitations:**

The motivation for the problem should be better presented, as well as the contributions. The authors make claims that are not fully justified throughout the text.

Additionally, the text needs an overall improvement, starting with the references and the figures.

More detailed examples of the effect of the frequency constraint and the effects of hyperparameters in the ablation studies would add strength to the paper.

More datasets and baselines would improve the paper's contributions.

**Quality:**

2

**Strengths And Weaknesses:**

# **Strengths**

**Sparse Views**

The paper directly tackles the challenging problem of reconstructing accurate surfaces from sparse-view images. The authors extend feed-forward networks to 2DGS and solve the inherent issues caused by these with frequency-based constraints on the Gaussian centers.

**Frequency-Based Constraint for Gaussian Surfels**

The introduction of Nyquist theorem-guided Gaussian surfel adaptations and feature aggregations is a technically sound approach to address the inherent geometric inaccuracies observed in prior feed-forward Gaussian radiance fields. The ablation study effectively validates the contribution of these modules to improved geometry and overall performance.

**Robustness and Detail**

The method is shown to deliver "superior global geometry and exhibits enhanced surface details," demonstrating improved global surface smoothness compared to competitors like UFORecon and FateGS.

___

# **Weaknesses**

**Motivation**

The related work section does not include a subsection on feed-forward networks, and in the introduction (2nd paragraph), it is not clear which papers present the issue and/or motivate the problem.

If the problem is the 2DGS-like structure in feed-forward networks, which networks are the authors referring to? Is the solution just an add-on to previous models? If so, the claim of a new feed-forward framework is not valid.

Additionally, Figure 2 does not help to clarify this. In line 46, there is no reference to the networks that exhibit the problem shown in Fig. 2(b), and there is no concrete example of the issue. For instance, an image generated by one of the methods would be a clearer representation of the problem.

**Efficiency**

The authors do not detail why their method achieves such a significant speed-up in inference. There is a comparison of efficiency against other methods, but how the speed was obtained is not clear. It seems that the speed comes from the fact that the authors use pixelSplat for their work's backbone, but they should not have claimed this as their own contribution. At least, it is not clear; there is nothing in their approach that accelerates inference.

Again, considering the efficiency of the method, in my opinion, the speed computations are not a fair comparison. The authors are comparing methods that require training from scratch, including the training time, against feed-forward methods that simply render the image. Even in the FateGS paper, the authors make that distinction.

**Results**

The authors only show results on the DTU dataset in the paper. They do not include results from BlendeMVS in the main paper—at least the quantitative results should have been included—and do not consider other, more complex datasets like Mip-NeRF 360.

Since the main motivation comes from solving a problem for feed-forward networks, the authors' evaluation is very limited. The authors follow the common practice of evaluating sparse views against explicit networks, while only comparing their method with FateGS, which is a feed-forward approach.

**Text and Figures**

References should not be grouped. They should be placed near where the paper is cited. Especially in the introduction and related work, the authors cite all the papers in the first sentence, and then the rest of the paragraph has no citations, making it hard to follow.

Some wording and examples are unusual, and there are a few typos. For example:
* "Verification" instead of "Proof" when proving the Nyquist Theorem Criterion.
* "Our key insight is that the failure to generate surface-aligned primitives is because the spatial frequency..." - The phrase "is because" is redundant.
* In Table 2: "Interference" should be "Inference."
* The graph in Figure 1 has an inverted y-axis.

---

> ### Author Rebuttal · Authors · 2025-07-30
>
> ## 1. Motivation
> ### 1.1 Inadequate introduction to Feed-Forward networks
> While we listed several feed-forward networks in line 38 of the Introduction, we acknowledge that recent progress in this area deserves more comprehensive coverage.
> Feed-forward models [1,2,3,4] learn powerful priors from large-scale datasets, so that 3D reconstruction and view synthesis can be achieved via a single feed-forward inference. These networks extract image features using an image encoder from multi-view input and regress pixel-aligned Gaussian primitives through depth and feature decoders. In the final version, we will add a dedicated subsection in Related Work to thoroughly introduce current feed-forward networks for generalizable 3D scene reconstruction. Specifically, we will analyze works including pixelSplat [1], MVSplat [2], HiSplat [3], and NoPoSplat [4], and so on.
>
> ### 1.2 Unclear presentation of the motivation
> The issue we address is a common limitation across previous feed-forward networks such as pixelSplat [1], MVSplat [2], HiSplat [3], and NoPoSplat [4], and the example in Figure 2 is from MVSplat [2]. We apologize for the unclear citation placement. We summarize our key motivations as follows:
>
> Existing feed-forward Gaussian networks reconstruct 3D scenes by predicting pixel-aligned Gaussian primitives, focusing on high-quality novel view synthesis.
> While these methods can achieve photorealistic renderings, we observe that the predicted Gaussians are parallel to the image plane rather than aligned with the actual surface geometry. As shown in Figure 2(a), we render the normals of MVSplat [2], which predicts similar normal directions for all Gaussians.
> For further demonstration, we evaluate current feed-forward networks [1,2,3,4] on DTU benchmarks and report the mean Chamfer distance, MSE loss and **the covariance of normal**.
> As shown in Table A, the high normal loss and small covariance of normal directions further confirm our findings.
>
> However, primitive orientation is critical for surface reconstruction. In Figure 2(b), the supervision is not sufficient to recover the attributes of such high-frequency pixel-aligned Gaussians given the sparsity of the observations.
> As evidenced in Table B, randomizing the predicted orientations of our network causes a 1.09dB PSNR drop while degrading reconstruction geometry on mean CD by 2.91.
> According to Nyquist's sampling theorem, we must operate at lower spatial frequencies to correctly predict Gaussian primitive orientations. This restriction in spatial sampling frequency for recovering Gaussian signals has also been noted in Mip-Splatting [5], which primarily addresses aliasing in multi-resolution rendering as a per-scene optimization framework.
>
>
> **Table A Evaluation of feed-forward networks on the DTU dataset**
> | | Normal Loss | Normal Covariance | Mean CD $\downarrow$ |
> |---|---|---|---|
> | pixelSplat [1]|  0.141 | 0.023 | 10.53 |
> | MVSplat [2]| 0.135 | 0.055 | 4.89 |
> | HiSplat [3]| 0.125 | 0.079 | 4.11 |
> | NoPoSplat [4]| 0.138 | 0.028 | 6.63 |
> | Ours | 0.060 | 0.540 |  1.12|
>
> **Table B Rendering quality on Re10k and surface reconstruction on DTU**
> | | PSNR $\uparrow$ |SSIM $\uparrow$| LPIPS $\downarrow$|Mean CD $\downarrow$|
> |---|---|---|---|---|
> |Ours| 26.39|0.869 | 0.128|1.12 |
> |Ours with random Gaussian orientations | 25.31| 0.825| 0.223| 4.03|
>
> ### 1.3 Is the solution just an add-on to previous models?
>
> While previous methods [1,2,3,4] directly regress Gaussians from the parameter prediction head, we introduce a circuit process to restricts the frequency by surfel adaptations and feature aggregation, which both are guided by the Nyquist sampling rates.
> We follow MVSplat [2] to construct our backbone image encoder, while our following pipeline is different from previous feed-foward networks, where Gaussian Splatting also serves as an intermediate self-parameter refinement mechanism for a circuit paradigm, which is different from a single-pass regression.
>
> Thanks for your suggestion. We will clarify our contribution on introducing the feed-forward model for Gaussian surface reconstruction from the perspective of Nyquisty Sampling Theorem.
>
> ## 2. Efficiency
> ### 2.1 Clarification on efficiency claims
>
> We acknowledge that the efficiency of our method benefits from the feed-forward architecture. However, integrating surface reconstruction effectively into feed-forward networks presents significant challenges: the orientations of Gaussian primitives cannot be correctly recovered due to insufficient spatial sampling frequency.
> To address this, we adopt surfel adaptation modules that enable each Gaussian primitive to acquire adequate geometric information, guided by the Nyquist sampling theorem, thereby achieving geometrically fine Gaussian radiance fields within the feed-forward framework.
>
> We will clarify that our method achieves superior efficiency because we solve the geometric orientation problem that has prevented effective utilization of feed-forward networks' inherent speed advantages. By overcoming this core limitation, we unlock the computational efficiency of feed-forward architectures while maintaining high-quality surface reconstruction.
>
> ### 2.2 Fairness of speed comparisons
>
> We acknowledge that the costly training process requires consideration. For a fairer comparison, we list the training and inference time alongside reconstruction performance on DTU benchmarks in Table C. While some methods with pre-training [1,2,7] provide fast inference time, other methods [6,10] take longer inference time for per-scene optimization.
> Our method shows superiority on reconstruction quality over both two kinds of methods.
> In practical applications, training can be done offline while users utilize the model online, making inference efficiency more critical than training time.
> For more comprehensive comparison, we will replace Table 2 by Table C, which takes both training and inference time into consideration.
>
> **Table C Comparison on efficiency and reconstruction quality.**
> | | Training Time| Inference Time | Mean CD $\downarrow$|
> |---|---|---|---|
> | FatesGS [6]| No need | 10 mins| 1.18 |
> | NeuSurf [10]|No need | 14 hours | 1.44|
> | UFORecon [7]| ~ 10 days | 60 s |1.91 |
> |pixelSplat [1]|~ 2 days | 0.104 s| 10.53|
> |MVSplat [2]|~ 2 days |0.044 s | 4.89|
> |Ours|~ 2 days | 0.997 s | 1.12|
>
> ## 3. Results
> ### 3.1 Quantitative results on BlendedMVS and Mip-NeRF 360 dataset:
> We fully acknowledge the importance of cross-dataset evaluation. We have provided qualitative BlendedMVS results in Figure 1 of our supplementary material.
> Following your recommendation, we conducted additional quantitative experiments on BlendedMVS following the settings in [8], as shown in Table D. Since Mip-NeRF 360 is designed for novel view synthesis without ground truth 3D meshes, we report novel view synthesis metrics on this dataset in Table E.
>
>
> **Table D Quantitative experiments on BlendedMVS**
> | ScanID| bear | clock | dog | jade | man | sculpture | stone | Mean CD $\downarrow$|
> |---|---|---|---|---|---|---|---|---|
> |FatesGS [6] |0.26 | 0.61| 0.83| 1.01|0.58 |0.47 | 0.40| 0.59|
> |2DGS [9] | 0.30| 1.06| 0.96| 1.16|0.61 | 0.52| 0.45| 0.73|
> |NeuSurf [10] |0.27 |0.93 | 0.83| 1.04| 0.57|0.50 |0.42 | 0.65  |
> |Ours |0.26 |0.53 | 0.80| 1.01|0.57 |0.46 |0.35 |  0.57 |
>
> **Table E Rendering quality on Mip-NeRF 360 dataset**
> | | PSNR $\uparrow$ |
> |---|---|
> |FatesGS [6] |  24.89|
> |2DGS [9] | 20.03|
> |NeuSurf [10] | 23.78|
> |Ours | 25.69|
>
> ### 3.2 Additional comparisons with feed-forward networks
> Thank you for this valuable suggestion! We agree that more feed-forward network comparisons would strengthen our evaluation. Note that FatesGS [6] is not a feed-forward method. Table A above presents comprehensive comparisons with other feed-forward networks.
>
> ## 4. Text & Figures
>
> ### 4.1 References should not be grouped.
> We will place references immediately after relevant statements. For example, in line 43, we will add citations directly after "current feed-forward networks" for improved clarity.
>
> ### 4.2 Some wording and examples are unusual.
> Thank you for the corrections! We will refine the vocabulary issues you identified. Regarding the inverted y-axis in Figure 1, this was an intentional design choice to enhance presentation clarity.
>
> ## 5. Paper Formatting Concerns
>
> ### 5.1 Supplementary material referenced but not present in the document.
> All referenced supplementary explanations are included in the supplementary materials package. If you encounter any specific formatting concerns, please let us know.
>
> ## References:
> > [1] Charatan D, Li S L, Tagliasacchi A, et al. pixelsplat: 3d gaussian splats from image pairs for scalable generalizable 3d reconstruction[C]\
> > [2] Chen Y, Xu H, Zheng C, et al. Mvsplat: Efficient 3d gaussian splatting from sparse multi-view images[C]\
> > [3] Tang S, Ye W, Ye P, et al. Hisplat: Hierarchical 3d gaussian splatting for generalizable sparse-view reconstruction[J]\
> > [4] Ye B, Liu S, Xu H, et al. No pose, no problem: Surprisingly simple 3d gaussian splats from sparse unposed images[J]\
> > [5] Yu Z, Chen A, Huang B, et al. Mip-splatting: Alias-free 3d gaussian splatting[C]\
> > [6] Huang H, Wu Y, Deng C, et al. FatesGS: Fast and accurate sparse-view surface reconstruction using gaussian splatting with depth-feature consistency[C]\
> > [7] Na Y, Kim W J, Han K B, et al. UfoRecon: Generalizable sparse-view surface reconstruction from arbitrary and unfavorable sets[C]\
> > [8] Zhang Y, Hu Z, Wu H, et al. Towards unbiased volume rendering of neural implicit surfaces with geometry priors[C]\
> > [9] Huang B, Yu Z, Chen A, et al. 2d gaussian splatting for geometrically accurate radiance fields[C]\
> > [10] Huang H, Wu Y, Zhou J, et al. NeuSurf: On-surface priors for neural surface reconstruction from sparse input views[C]

---

> > ### Comment · Area_Chair_nXww · 2025-08-03
> >
> > Reviewer Wk3j, did the rebuttal address your concerns? Do you have any further questions or comments for the authors?

---

> > ### Comment · Reviewer_Wk3j · 2025-08-06
> >
> > I would like to thank the authors for their extensive rebuttal.
> >
> > **Motivation**: The authors have clarified my concerns on this topic. The addition of Table A, an updated version of Figure 2, and the accompanying comments are essential additions that must be included in the final paper. These changes help clarify the motivation behind the solution and have addressed my main concerns.
> >
> > **Efficiency**: The efficiency claim appears to be validated by Table B. However, I still have two main concerns:
> >
> > 1) The way efficiency is currently presented in the paper is misleading and must be updated as promised in the rebuttal (in response to Q2.1). This is a critical point, as the current presentation appears to invalidate Figure 1 and one of the paper's main contributions. For instance, the comparison with FateGS needs to be presented more accurately. Although its performance is slightly worse, FateGS does not require training and obtains acceptable results in approximately 10 minutes. In contrast, the proposed solution is refined for two days before inference can be performed in about one second. This is a crucial distinction, especially when the final CD and PSNR metrics are nearly identical to those of FateGS across three different datasets.
> >
> > 2) I still do not understand how, exactly, solving for orientations enables efficiency in a feed-forward network. What is the precise reason for this efficiency gain? Does it reduce the number of Gaussians? How does the orientation of the surfels in inference allows a faster speed-up? In line 142, the authors refer to Gaussians that fail to meet the Nyquist frequency criteria but that sampling is used during training. This point was not clarified in the rebuttal either.
> >
> > **Results**: The additional results strengthen the paper. Nonetheless, they also highlight the vulnerability of the efficiency claim, as discussed above.
> >
> > **Text, Figures, and Formatting**: The authors promised to refine the text and address my issues with the references. Their clarification regarding which baseline methods are feed-forward networks helped to resolve some of my concerns. Also, I apologize for the sup. material comment. I will retract it from the official comment.

---

> > > ### Author Response · Authors · 2025-08-08
> > >
> > > We sincerely appreciate your valuable feedback. We acknowledge that our current presentation of efficiency requires clarification and will implement the following refinements:
> > >
> > > ## Addressing the Misleading Efficiency Presentation
> > >
> > > We agree that the current efficiency claims require more nuanced presentation. Here are our planned revisions:
> > >
> > > **Modification of Figure 1:** We recognize that Figure 1(a) creates confusion by comparing our feed-forward approach directly with optimization-based methods (2DGS [1], NeuSurf [2]), which may misrepresent our core contribution. We will implement two key modifications:
> > > 1. **Visual Comparison Update:** The upper row will be replaced with detailed comparisons among feed-forward methods, specifically showcasing depth and normal maps from existing approaches. This will clearly demonstrate our method's unique capability in producing geometrically accurate Gaussian radiance fields within the feed-forward paradigm.
> > > 2. **Pre-training Transparency:** We will add a pre-training stage visualization indicating the 2-day training period, ensuring full transparency about the computational requirements.
> > >
> > > **Revision of Introduction:** Lines 65-66 will be revised to: "Experimental results demonstrate the effectiveness and generalizability of our method. After sufficient pre-training, SurfaceSplat generates surface-aligned Gaussian radiance fields with accurate geometry. Compared with existing reconstruction methods, SurfaceSplat achieves faster inference performance."
> > >
> > > **Expanded Experimental Analysis:** Table 1 will include pre-training time to provide complete efficiency context. The analysis (lines 222-226) will be expanded as follows:
> > > 1. **Inference Performance:** While our method achieves the fastest runtime among surface reconstruction approaches, this efficiency stems from successfully adapting feed-forward networks through our orientation solution.
> > > 2. **Training Investment:** The fast runtime requires substantial pre-training investment, which we acknowledge as an important consideration.
> > > 3. **Quality-Speed Trade-off:** Despite the pre-training requirement, our method achieves superior reconstruction quality compared to all existing approaches.

---

> > > > ### Author Response · Authors · 2025-08-08
> > > >
> > > > ## Clarifying the Efficiency Gain
> > > >
> > > > **Clarifying the Efficiency Gain:** We clarify that our orientation solution does not provide efficiency gains relative to other feed-forward networks [3,4]. Our approach does not accelerate the feed-forward process itself, nor does it require reducing the number of Gaussians. While to our best knowledge, most existing surface reconstruction methods are optimization-based, our key motivation is to apply feed-forward networks to the domain of surface reconstruction, thereby achieving stronger efficiency and generalizability compared with optimization-based methods.
> > > >
> > > > To achieve this, the main challenge is that existing feed-forward approaches fundamentally fail to recover accurate surface geometry. As demonstrated in Rebuttal Table C, while methods like MVSplat [3] and pixelSplat [4] achieve extremely fast inference (~0.1 seconds), they cannot represent precise surface information—the geometric orientation problem prevents effective utilization of feed-forward networks for surface reconstruction tasks.
> > > >
> > > > Therefore, we design Nyquist-guided surfel adaptation and feature aggregation modules to solve this challenge. Our contribution enables feed-forward networks to successfully perform surface reconstruction by leveraging large-scale pre-training to accumulate extensive scene priors, which allows rapid surface extraction without per-scene optimization [1,2]. This results in significant efficiency improvements over optimization-based surface reconstruction methods (Table 2 in our paper) while maintaining only a modest increase in inference time (~1 second) compared to feed-forward methods that cannot perform surface reconstruction.
> > > >
> > > > In essence, we bridge the gap between the efficiency of feed-forward networks and the accuracy requirements of surface reconstruction, making feed-forward surface reconstruction feasible for the first time.
> > > >
> > > > **Nyquist Sampling Clarification:** Regarding line 142, we do not prune Gaussians failing Nyquist criteria. Instead, our surfel adaptation and feature aggregation modules ensure all Gaussians satisfy sampling requirements through:
> > > > 1. **Surfel Adaptation Module:** Designs low-pass filters for each Gaussian based on spatial sampling frequency (Section 3.1.2) and Gaussian surfel spatial frequency (Section 3.1.3).
> > > > 2. **Feature Aggregation Module:** Projects adapted Gaussians onto input views to identify image regions relevant for orientation learning, then applies cross-attention to refine features and re-predict geometric attributes (Section 3.2.2).
> > > >
> > > > This design ensures that all Gaussians maintain geometric accuracy for surface reconstruction while achieving efficiency through pre-trained priors rather than iterative per-scene optimization.
> > > >
> > > > ## References
> > > > > [1] Huang B, Yu Z, Chen A, et al. 2d gaussian splatting for geometrically accurate radiance fields[C]//ACM SIGGRAPH 2024 conference papers. 2024: 1-11.\
> > > > > [2] Huang H, Wu Y, Zhou J, et al. NeuSurf: On-surface priors for neural surface reconstruction from sparse input views[C]//Proceedings of the AAAI conference on artificial intelligence. 2024, 38(3): 2312-2320.\
> > > > > [3] Chen Y, Xu H, Zheng C, et al. Mvsplat: Efficient 3d gaussian splatting from sparse multi-view images[C]//European Conference on Computer Vision. Cham: Springer Nature Switzerland, 2024: 370-386.\
> > > > > [4] Charatan D, Li S L, Tagliasacchi A, et al. pixelsplat: 3d gaussian splats from image pairs for scalable generalizable 3d reconstruction[C]//Proceedings of the IEEE/CVF conference on computer vision and pattern recognition. 2024: 19457-19467.

---

### Decision · Program_Chairs · 2025-09-17

**Decision:**

Accept (poster)

**Comment:**

This paper introduces SurfaceSplat, a feed-forward approach for surface reconstruction from images using 2D Gaussian surfels. The key insight is that 2D surfels are often incorrectly oriented parallel to the image plane due to a violation of the Nyquist sampling theorem: the spatial frequencies of pixel-aligned surfels are too high for the given sampling rate. The paper proposes a practical solution that allows for both fast inference times and superior geometry compared to state-of-the-art feed-forward methods.
During the review process, reviewers were satisfied with the novel Nyquist theorem-guided Gaussian surfel adaptations and the method's fast and accurate results.
However, the reviewers also raised several significant concerns, including:
1. A missing discussion of related feed-forward methods.
2. An unclear efficiency-accuracy trade-off when compared to other optimization-based and feed-forward models.
3. A lack of experiments on datasets other than DTU.
4. An unclear presentation of the Nyquist sampling issue and its theoretical justification.
5. Missing implementation details.

During the rebuttal and discussion phase, the authors thoroughly addressed these concerns, convincing three out of four reviewers to be in favor of accepting the paper. Since the paper provides an insightful, Nyquist Theorem-inspired analysis of a key issue with feed-forward 2D Gaussian surfels and presents an effective solution that significantly improves upon previous work, I recommend this paper for acceptance. I expect the authors to address all concerns raised by the reviewers in the final version by incorporating the answers and results provided during the review process.